



# GNSS-RO Residual Ionospheric Error (RIE): A New Method and Assessment

Dong L. Wu[1], Valery A. Yudin[2], Kyu-Myong Kim[1], Mohar Chattopadhyay[3], Lawrence Coy[3], Ruth S. Lieberman[1], C. C. Jude H. Salinas[4], Jae N. Lee[5], Jie Gong[1], and Guiping Liu[1]

[1]NASA Goddard Space Flight Center, Greenbelt, MD 20771, USA
[2]Department of Physics, The Catholic University of America, Washington, DC 20064, USA
[3]Science Systems and Applications Inc., Lanham, MD 20706, USA
[4]GESTAR-II, University of Maryland Baltimore County, MD 21250, USA
[5]JCET, University of Maryland Baltimore County, MD 21250, USA

*Correspondence to*: Dong.L.Wu (dong.l.wu@nasa.gov)

**Abstract.**

GNSS radio occultation (RO) observations play an increasingly important role in monitoring climate changes and numerical weather forecasts in the upper troposphere and stratosphere. The magnitudes of the RO bending angle are small at these altitudes, and therefore residual ionospheric error (RIE) is critical to accurately retrieve atmospheric temperature and

refractivity. The latter represent the state variables of the weather and climate models. RIEs remain poorly characterized in terms of the global geographical distribution and its variations with the local time and altitude influenced by the solar cycle and solar-geomagnetic disturbances. In this study we developed a new method to determine RIE from the RO excess phase measurement on a profile-by-profile basis. The method, called $\phi_{ex}$-gradient method, is self-sufficient and based on the vertical derivative of the RO excess phase ($\phi_{ex}$) profile, which can be applied to individual RO bending angle observations

for RIE correction. In addition to the RIE in bending angle measurements, RIEs are found in the RO $\phi_{ex}$ measurements in the upper atmosphere where an exponential dependence is expected and in small-scale temperature variance of the RO retrieval. We found that the RIE values derived from the $\phi_{ex}$-gradient method can be both positive and negative, which is fundamentally different from the $\kappa$-method that produces only the positive RIE values. The new algorithm reveals a latitude-dependent diurnal variation with a larger daytime negative RIE (up to ~3 µrad) in the tropics and subtropics. Based on the

observed RIE climatology, a local-time dependent RIE representation is used to evaluate its impacts on reanalysis data. We examined these impacts by comparing the data from the Goddard Earth Observing System (GEOS) data assimilation (DA) system with and without the RIE. The RIF impact on GEOS DA temperature is mainly confined to the polar regions of stratosphere. Between 10 hPa and 1 hPa the temperature differences are ~1K and exceed ~3-4 K in some cases. These results further highlight the need for RO RIE correction in the modern DA systems.

Keywords: residual ionospheric error, GNSS radio occultation, bending angle, data assimilation, reanalysis, climate data record, upper stratosphere, long-term variation









# 1      Introduction

The global navigation satellite system (GNSS) radio occultation (RO) data have been assimilated at most of the numerical weather prediction (NWP) centers for global and regional analysis/reanalysis with a RO processing package or ROPP [Culverwell et al. 2015]. Because of the high accuracy of RO measurements in the Upper Troposphere and Lower Stratosphere (UT/LS), GNSS-RO data have become a valuable source of information in data assimilation (DA) systems for climate and weather predictions and applications [Foelsche et al., 2011; Kursinski et al., 1997]. Assimilating GNSS-RO

vertical profiles of the bending angle ($\alpha$ or BA) was found to have significantly positive impacts on weather prediction skills [Poli et al., 2010; Cucurull et al., 2013], both directly through improvements of temperature and humidity forecasts in the UT/LS, and indirectly by utilizing the RO temperature and humidity analyses as the 'true' profiles for the radiance bias correction. The recent Observing System Simulation Experiments (OSSEs) seem suggest that the global coverage and growth of GNSS-RO data will continue improve the forecast skills of the current DA systems without the saturation for the

positive impacts of RO observations [Harnisch et al., 2013; Prive et al., 2022].

      However, the GNSS-RO data infusion requires a key assumption about the $\alpha$ measurements in which ionospheric contributions can be fully removed by using the sounding from two L-band frequencies (a.k.a, L1 and L2) [Vorob'ev and Krasil'nikova, 1994; Culverwell et al., 2015]. In other words, Residual Ionospheric Errors (RIEs) after removal of ionospheric contributions by the dual-frequency (L1/L2) scheme of Vorob'ev and Krasil'nikova [1994] would be small and

negligible in the DA of $\alpha$ measurements. Yet, recent studies have found that the RIEs may not be as small as previously thought and can have a significant impact on the DA of RO observations. For example, Danzer et al. [2013] highlighted an unrealistic solar cycle variation in the mean neutral atmospheric temperature. Because the GNSS-RO data have been increasingly assimilated in global analysis and reanalysis systems for climate records, it remains unclear what is the amplitude of RIE-induced bias and variability in the neutral atmospheric variables. The variance of these variables can be as

important as their mean values, because the analysis and reanalysis data have been widely used to study atmospheric planetary and gravity waves.

      Identifying the RIE sources and amplitudes and developing the RIE correction methods remain as an active research topic. Higher-order ionospheric refractive index and frequency-dependent pathway difference are considered as the leading causes of the RIE. Without the dual-frequency first order correction, the ionospheric bending can induce in a pointing error

of ~100 m in $h_t$ and +/- 0.02° in BA [Hajj and Romans, 1998]. Higher-order contributions not removed by the linear combination of L1 and L2 measurements may depend on several factors. Most important among them are ionospheric structure [Ladreiter and Kirchengast 1996; Syndergaard 2000; Mannucci et al., 2011], magnetic field and electron density ($Ne$) [Hartmann and Leitinger, 1984; Brunner and Gu, 1991; Morton et al., 2009; Hogan and Jakowski, 2011], radio wave propagation path [Coleman and Forte, 2017], and horizontal inhomogeneity [Syndergaard and Kirchengast, 2022].

Depending on the mechanism that can contribute to RIE, the magnitudes of RIEs can vary from $10^{-8}$ rad to $10^{-6}$ rad in $\alpha$. For climate studies that relied on the reanalysis data, it is imperative that the GNSS-RO RIEs must be clearly characterized and





removed, since climate change signals are often small and comparable to the RIE amplitude [Ringer and Healy, 2008; Gleisner et al., 2022].

Several methods have been proposed to correct RIE impacts on the $\alpha$ measurements before they are assimilated [Syndergaard 2000; Gorbunov, 2002; Healy and Culverwell, 2015; Zeng et al., 2016; Angling et al., 2018; Liu et al., 2020; Danzer et al., 2021]. Syndergaard [2000] emphasized the ionospheric E-layer impacts where the L1 and L2 may propagate through slightly different paths due to sharp vertical gradients at the lower ionosphere such as sporadic-E. Such path differences can result in an error of as large as ~1 m in iono-free or atmospheric excessive phase ($\phi_{ex}$) measurements in the E-region or -0.3 μrad in $\alpha$ at $h_t$ =60 km, but it gradually decreases with $h_t$. Gorbunov [2002] developed an optimal estimation method, by balancing between the $\alpha$ measurement error and its climatology at high $h_t$, to reduce RIE impacts on the lower atmosphere. Healy and Culverwell [2015] introduced a so-called $\kappa$-method to remove high-order RIE contributions to $\alpha$, which is proportional to the squared difference between L1 BA ($\alpha_1$) and L2 BA $\alpha_2$) or $\alpha_{RIE}(h_t) = \kappa \cdot (\alpha_1 - \alpha_2)^2$. The $\kappa$ profile is estimated using realistic ionospheric *Ne* profiles and has a typical value between 10-20 rad$^{-1}$ to correct only RIEs of negative values [Healy and Culverwell, 2015; Angling et al., 2018]. To improve the open-loop (OL) tracking of L2C signal, Zeng et al. [2016] applied an empirical method for the ionospheric correction and extrapolate the $\alpha_1 - \alpha_2$ profile down to a very low $h_t$. Their extrapolation approach from fitting high-$h_t$ $\alpha_1 - \alpha_2$ is similar to the $\kappa$-method, except that some large (between -10 and -30 μrad) $\alpha_1 - \alpha_2$ values were found at $h_t$ =60 km. Angling et al. [2018] showed a similar amplitude (-10 μrad) of the $\alpha_1 - \alpha_2$ at $h_t$ =60 km and further extended the $\kappa$ method by providing a global model as a function of solar zenith angle ($\chi$) and solar cycle characterized by the solar radio flux at 10.7 cm (F10.7). The $\kappa$ model predicts a lower value in day and during a higher F10.7. On the other hand, Liu et al. [2020] had estimated the RIE-induced $\alpha_1 - \alpha_2$ with a different method, showing small values less than 0.1 μrad at $h_t$ =60 km. Danzer et al. [2020] implemented and evaluated the $\kappa$-model for RIE correction with the European Center for Medium-range Weather Forecast reanalyses (ERA-Interim, Dee et al., 2011; and ERA5, Hersbach et al., 2020), reporting warming (0.2 – 2 K) effects at 40-45 km. Using a 3D ray tracing technique, Li et al. [2020] found that the simulated RIE can be both positive and negative on the order of ±0.1 μrad. In summary, the current and recent studies display considerable differences in the estimated magnitudes and morphologies of RIEs. As a result, it remains unclear what spatiotemporal distribution is the correct representation of these RIEs and how these errors could impact on the assimilated data, in terms of local time and solar cycle variations, when the RIE-prone RO data are injected to DA systems.

In this study we developed a new method for RIE estimation, using the vertical gradient of RO $\phi_{ex}$ profile calculated at high $h_t$, hereinafter referred to as the $\phi_{ex}$-gradient method. We show that this vertical gradient is directly related to the RIE-induced $\alpha_1 - \alpha_2$ difference, and the RIE value determined at high $h_t$ can be extrapolated to the $\alpha$ measurements in the low $h_t$ domain. The analysis of estimated $\alpha_1 - \alpha_2$ values reveals a different morphology in terms of diurnal cycle, latitudinal variability and solar cycle dependence. Impacts of the diurnal and latitudinal variations of RIE specified by the $\phi_{ex}$-gradient



method is assessed by performing the DA experiments with and without the RO RIE in the Goddard Earth Observing System
for Instrument Teams (GEOS-IT).

## 2      GNSS-RO Data

### 2.1      Atmospheric Bending Angle ($\alpha$) and Excess Phase ($\phi_{ex}$)

Bending of the RO ray path occurs where there exists a vertical gradient in refractive index *n*, which can be from the
ionosphere and the neutral atmosphere, and the bending angle is given by

$$\alpha = -2a \int_a^\infty \frac{1}{n\sqrt{n^2r^2-a^2}} \left(\frac{dn}{dr}\right) dr \qquad (1)$$

where *a* is impact parameter and *r* is radius from the Earth center. If *dn/dr* <0, as in the neutral atmosphere, the propagation
ray is bended down towards the Earth ($\alpha$ <0). In the ionosphere the bending can be both upwards and downwards. In the top
of ionosphere, where *dn/dr* >0, the propagation ray tends to be bended upwards, whereas in the bottom of ionosphere, where
*dn/dr* <0, the ray is bended downwards by the vertical gradient of Ne.

135       The ionospheric bending between GNSS transmitter and LEO (low Earth orbit) receiver depends on transmitter's
radio wave frequency while the neutral atmospheric bending is independent of the frequency. The first-order ionospheric
bending effect can be removed using a linear combination of GNSS-RO processing [Vorob'ev and Krasil'nikova, 1994] as
follows

$$\alpha = f_1^2/(f_1^2 - f_2^2) \cdot \alpha_1 - f_2^2/(f_1^2 - f_2^2) \cdot \alpha_2 \qquad (2)$$

where $f_1$ and $f_2$ are L1 and L2 frequencies, and $f_1^2/(f_1^2 - f_2^2) = 2.5457$ and $f_2^2/(f_1^2 - f_2^2) = 1.5457$. In the absence of
ionospheric bending, $\alpha = \alpha_1 = \alpha_2$, which is denoted by $\alpha_c$ as the correct value. In the case wehre the ionospheric bending
effect is not completely removed by Eq.(2), an RIE exists in the $\alpha$ measurement, mathematically

$$\alpha = \alpha_c + \alpha_{RIE} \qquad (3)$$

In a simplified wave propagation model, Melbourne (2004) obtained a first-order linear relation between $\alpha$ and the
excess Doppler (i.e., $\phi_{ex}$ derivative with respect to time),

$$\alpha \cong -\frac{d\phi_{ex}}{dt} \cdot \frac{1}{V_\perp} \qquad (4)$$

where $V_\perp$ is the LEO motion perpendicular to its line of sight (LOS) to the GNSS transmitter. The atmospheric $\phi_{ex}$ can be
obtained from L1 ($\phi_{exL1}$) and L2 ($\phi_{exL2}$) phase measurements with the linear combination similar to Eq.(2)

$$\phi_{ex} = f_1^2/(f_1^2 - f_2^2) \cdot \phi_{exL1} - f_2^2/(f_1^2 - f_2^2) \cdot \phi_{exL2} \qquad (5)$$

150       For a rising/setting occultation, $V_\perp$ is the ascending/descending rate of RO sampling with respect to $h_t$, or the
GNSS–LEO straight line height (SLH), which yields $V_\perp \cong dh_t/dt$. In the upper atmosphere $V_\perp$ is typically ~2 km/s.
Substituting this $V_\perp$-$h_t$ relation into Eq.(4), we have

$$\alpha \cong -d\phi_{ex}/dh_t \qquad (6)$$





In the upper atmosphere where there is little atmospheric bending (i.e., $\alpha_c \approx 0$), a significant value that is not zero in

$d\phi_{ex}/dh_t$ indicates the existence of $\alpha_{RIE}$, which can be both positive and negative. Thus, Eq.(6) is used as a theoretical

basis in this study to derive $\alpha_{RIE}$ from the GNSS-RO Level-1B data.

## 2.2    RIE and Detection Method

For accurate estimation of the climate temperature trends from the GNSS-RO, it is very important to introduce the

adequate algorithm to correct RIE in the observed vertical $\alpha$ profiles. At high altitudes, where magnitudes of RIEs may equal

or/and exceed the background $\alpha$ values, the observed $\alpha$ noise significantly vary from profile to profile. As shown in Figs.1-3,

the oscillatory nature of $\alpha$ profile between 60 and 80 km preclude it from utilizing the $\alpha$ profile to reliably determine the

RIE. In the case in Fig.1, a positive bias might exist in the mean $\alpha$ value at 60-80 km, whereas it is not a clear case in Fig.2

where an E-layer may have contaminated the profile. In addition, the atmospheric bending remains non-negligible at 60 km.

Thus, in this study we focus on the estimation of RIE using the RO data at heights above 65 km.

Although ionosphere-induced $\alpha$ oscillations in $\alpha_1$ and $\alpha_2$ are mostly removed by Eq.(2), the ionospheric residuals

are still significant and can occur at different altitudes with different oscillatory behaviors [Figs. 1-3]. For example, the small

$\alpha$ bump at ~60 km in Fig.2b is not obviously associated with the $\alpha_1$ and $\alpha_2$ oscillations, while the larger negative one near

70-74 km are correlated with them. These residuals can be readily induced in the $\phi_{ex}$ measurements and their height

derivatives. Another example is a step jump of the $\phi_{ex}$ profile at ~70 km in Fig.3, which results in a large and sharp spike in

$\alpha$. Despite the high (100-Hz) sampling rate of COSMIC-2 GNSS-RO, which helps to remove more ionospheric effects than

the data from a lower sampling, these RIE signatures can still be seen in the in $\alpha$ and $\phi_{ex}$ profiles. On one hand, the sharp

spikes in the $\alpha$ profile like one in Figs.2-3 may not play a significant role in the lower atmospheric retrievals, because these

RIEs tend to be confined near the spike altitude. On the other hand, a spike in the $\phi_{ex}$ profile from the sporadic-E (Es) event

is superimposed on a systematic slope $(d\phi_{ex}/dh_t)$ or $\alpha$ bias, which should be considered as an RIE as seen in Fig.4. The

RIE in Fig.4 exhibits an extended slope that appears to originate from the ionosphere above ~100 km (Fig 4c). This slope can

have an extended impact on the $\alpha$ profile below 100 km.

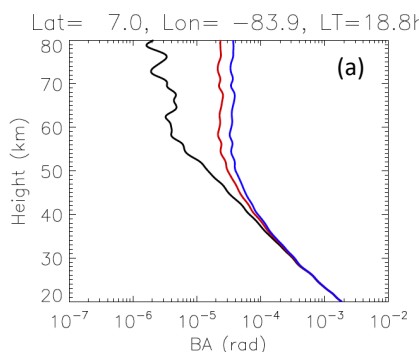
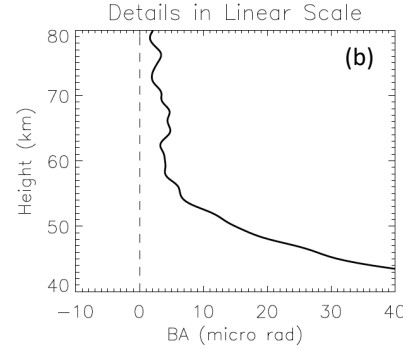
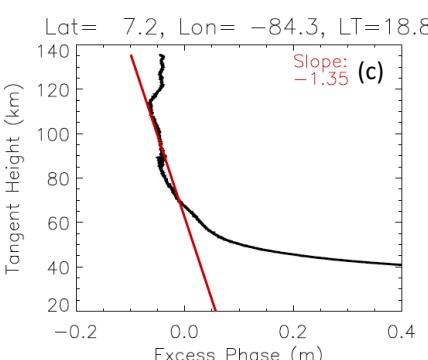





Fig.1. An example of COSMIC-2 atmospheric $\alpha$ profiles in (a) logarithmic and (b) linear scale, and (c) the corresponding $\phi_{ex}$ profile from January 1, 2022. The red and blue profiles in (a) are L1 and L2 $\alpha$ respectively. The dashed line in (b) denotes no RIE if the $\alpha$ average at 60-80 km is zero. The red line in (c) is a linear fit to the $\alpha$ data between 65 and 120 km, of which the slope is $d\phi_{ex}/dh_t$.

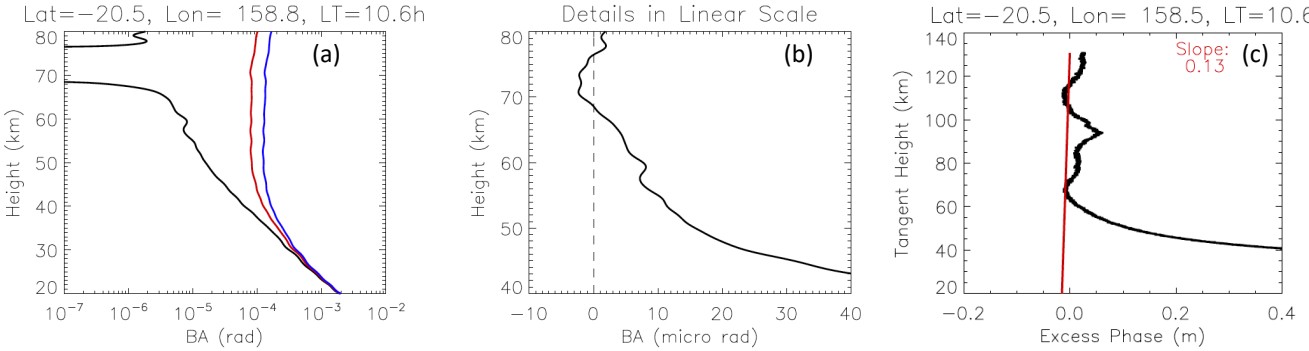

Fig.2. As in Fig.1 but for another example where an ionospheric E-layer is present in the $\phi_{ex}$ profile.

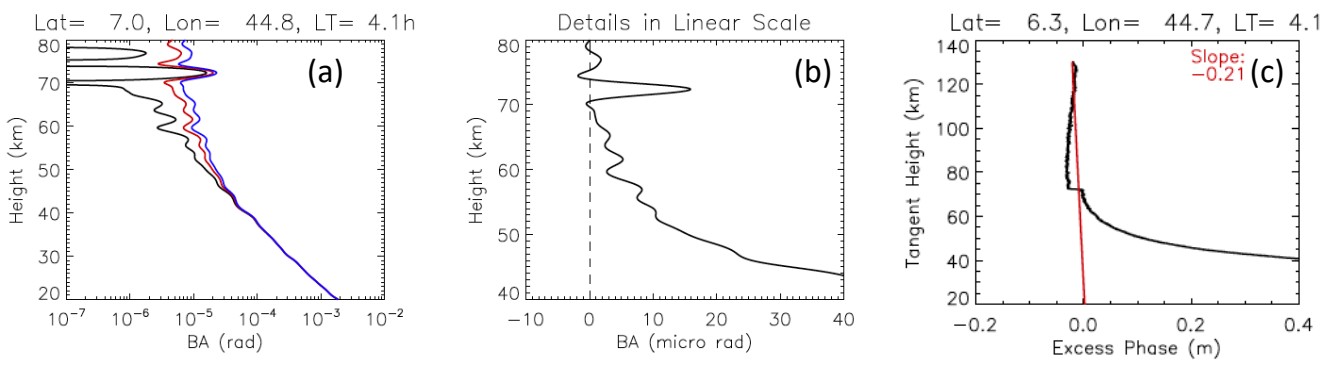


Fig.3. As in Fig.1 but for a case with a sharp jump in the $\phi_{ex}$ profile near 70 km and a moderate derivative $d\phi_{ex}/dh_t$.

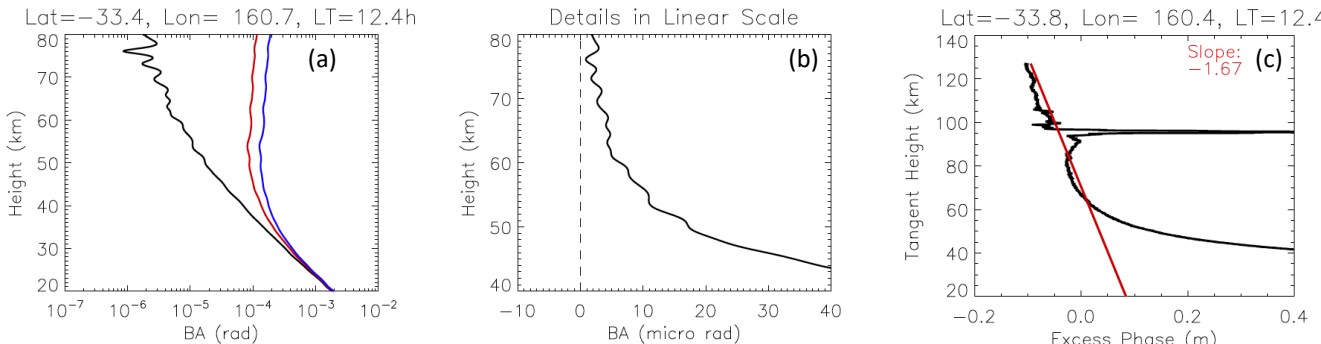

Fig.4. As in Fig.1 but for a case with a sharp spike in the $\phi_{ex}$ profile near 95 km and a significant derivative $d\phi_{ex}/dh_t$.



Thus, it is important to develop a method to overcome the noisy/oscillatory nature of $\alpha$ profiles and estimate the RIE that may have an impact on the temperature retrieval in the upper troposphere and stratosphere. A robust RIE algorithm needs to demonstrate the following capabilities: (i) to adequately handle short/sharp spikes in the E-region; (ii) to be self-sufficient in determining the RIE for each individual $\alpha$ profile, regardless of the ionospheric conditions; and (iii) to be able to make a RIE estimate in the presence of large noise. Using the relationship between the $\alpha$ and $d\phi_{ex}/dh_t$ as described by

[Eq.6], we introduce a RIE correction scheme for the $\alpha$ profile, which can be implemented at Level-1B $\phi_{ex}$ (excess phase) processing. There are several advantages to use the Level-1B data for the RIE correction. They are as follows:

1.   As a low level product, $\phi_{ex}$ is a more fundamental measurement than $\alpha$. The inversion from $\phi_{ex}$ to $\alpha$ profile in RO data reduction can introduce additional noise. The $\phi_{ex}$ data do not contain additional smoothing and a priori constraints employed by the inversion algorithms that could vary between software developers and versions.

2.   For the RIE estimation, we can carry out a simple least-squared linear fit to the iono-free $\phi_{ex}$ profile to derive $d\phi_{ex}/dh_t$ without applying the inversion. For a reliable fit, quality control can be applied directly to the Level-1B data.

3.   $\phi_{ex}$ contains the information about the ionosphere that was not corrected by the L1 ($\phi_{ex_{L1}}$) and L2 ($\phi_{ex_{L2}}$) measurements. The Level-1B high-rate sampling is retained useful insights and allow further investigations on the

cause(s) of RIEs.

To estimate the RIE for $\alpha$, we carry out a linear fit to the iono-free $\phi_{ex}$ data at $h_t > 65$ km for given profiles of RO data, i.e.

$$\phi_{ex} = -\Delta\alpha \cdot h_t + \phi_0 \qquad\qquad h_t > 65 \text{ km} \qquad\qquad (7)$$

where $\phi_0$ is the constant from the fitting and the slope $\Delta\alpha \equiv -d\phi_{ex}/dh_t$. $\Delta\alpha$ can be both positive and negative. If there is no RIE, $\Delta\alpha = 0$. We attribute $\Delta\alpha \neq 0$ as the RIE for the corresponding $\alpha$ profile. The 65 km altitude cutoff in the $\Delta\alpha$ calculation is to ensure that the fitting will not be influenced by the neutral atmospheric bending. To compare with the RIE estimated from the $\kappa$-method, we simply multiply $(\Delta\alpha_1 - \Delta\alpha_2)^2$ with $\kappa$, where $\Delta\alpha_1$ and $\Delta\alpha_2$ are respectively for $\phi_{ex_{L1}}$ and $\phi_{ex_{L2}}$. As shown and discussed in the following section, there are important differences in the RIE climatology derived from

these methods for RIEs. Note that Eq.7 does not rely on any auxiliary data/model such as the international reference ionosphere (IRI), nor assumptions about the spherical symmetry of electron density (Ne) profile. As shown in Mannucci et al. [2011] and Coleman and Forte [2017], the spherical symmetry assumption can become problematic for the RIE evaluation in the presence of complex ionospheric structures and small-scale variability of the ionosphere.

Quality control (QC) on the $\phi_{ex}$ data is required to ensure the fitting yields a reasonable $\Delta\alpha$. Table 1 summaries the

QC flags and procedures applied to GNSS-RO data in the RIE estimation. Because of complex impacts of ionospheric variabilities on GNSS-RO sounding, the $\phi_{ex}$ data processing algorithm needs to have adequate treatments of large $\phi_{ex}$ oscillations and non-monotonic profiles with multiple extremes at altitudes > 65 km. These extraordinary $\phi_{ex}$ profiles





include multiple layers with different $\phi_{ex}$ slopes in between, large jumps in the measurement, noisy profiles due to ionospheric scintillations, and disturbances from Es layers. With the QC#1 and QC#2 criteria in Table 1, we ensure that the

$\phi_{ex}$ profile has a sufficient number of samples and good signal-to-noise ratio (SNR) at 60-120 km to allow a useful fit from Eq.7. Since the GNSS-RO L1 ($\phi_{ex_{L1}}$) and L2 ($\phi_{ex_{L2}}$) measurements do not have absolute calibration, they are simply initialized from the top of each RO profile. As a result, the derived $\phi_{ex}$ values at 60-120 km are not too far from zero as shown in Figs.1-4. However, profiles with very large $\phi_{ex}$ values and large standard deviation about its mean do exist. For this consideration, QC#3 and OC#4 are applied to the $\phi_{ex}$ data prior to the fitting. As noted above, Eq.7 can handle a

constant offset in the $\phi_{ex}$ data after screened by QC#3, but QC#4 helps to minimize the impacts from data spikes (e.g., Es) and E-layer residuals (e.g., Fig.2). In some algorithm experiments (see the discussion section), the $\phi_{ex}$ profile is required to have a top reaching the certain height limits. Lastly, the $\phi_{ex}$ data may have a large gap in the profile, which could also yield a problematic fit from Eq.7 and should be excluded (QC#6). In this study we are not interested in very large $\Delta\alpha$ values that are greater than 2000 μrad (QC#7).


Table 1. QC in $\phi_{ex}$ Data Processing for $\Delta\alpha$

| QC# | Description | Threshold |
|---|---|---|
| 1 | Ensure that number of $\phi_{ex}$ at 60-120 km is sufficient | N > 200 |
| 2 | Check if mean RO signal at 60-120 km is too weak | L1 SNR > 100 |
| 3 | Check if mean $\phi_{ex}$ ($\overline{\phi_{ex}}$) at 60-120 km is too large | $\left|\overline{\phi_{ex}}\right| < 30$ m |
| 4* | Exclude large deviations from the mean ($\phi_{ex} - \overline{\phi_{ex}}$) at $h_t > 65$ km | $\left|\phi_{ex} - \overline{\phi_{ex}}\right| < 0.05$ m |
| 5* | Check the top $h_t$ of RO profile | $h_t > 80$ km, 120 km, 170 km |
| 7 | Avoid profiles with large $h_t$ gaps at 60-120 km | $\left|dh_t\right| < 2$ km |
| 8 | Retain only realistic $\Delta\alpha$ values | $\left|\Delta\alpha\right| < 2000$ μrad |

Note: *Thresholds in the QC#4 and QC#5 require further tests to achieve optical results.

## 3.  RIEs from $\phi_{ex}$-Gradient Method

### 3.1  $\Delta\alpha$ Morphology

Statistical properties of the $\Delta\alpha$ estimated from the slope ($d\phi_{ex}/dh_t$) at high altitudes vary with local time, latitude, season, and solar activity, and they may also depend on RO receiver type. To examine the probability distribution from different RO receivers, we first derive $\Delta\alpha$ without imposing QC#4 in Table 1 and aggregate the monthly $\Delta\alpha$ data in terms of a normalized probability density function (PDF) as a function of latitude separately for day and night. Because of large differences in sampling number, the PDFs are normalized at each latitude bin to its peak value. Figs. 5-8 show the results

from COSMIC-1 (Jan 2013), COSMIC-2 (Jan 2022), Spire (Jan 2022), MetOp-B (Jan 2023) and FY3-E (Jan 2023). To





compare these statistics for a similar environment, we chose the year of 2013 and 2022-2023 when there were high solar activities. The ionosphere in January has a generally higher Ne in the southern hemisphere (SH) from more solar illumination. For these $\Delta\alpha$ comparisons, the top height of GNSS-RO profiles (QC#5) is required to reach 120 km, which has not been common in the regular operation of most meteorological satellites (e.g., MetOp and FY3) until recently. Since early

2022 MetOp-B/C have begun to acquire GNSS-RO profiles above 120 km during the routine operation with the equator-crossing-time (ECT) of (8-10h and 20-22h), which allows the comparison with the COSMIC observations. Both COSMIC-1 and COSMIC-2 constellations, as well as Spire, have a coverage of all local times over a period of one month. The new GNSS receivers on FY3-E (March 2022-present) have been providing the RO sampling with a top above 120 km with the ECTs of (4-6h and 16-18h).

The comparison between COSMIC-1 and COSMIC-2 $\Delta\alpha$ PDFs [Figs.5-6] shows a slightly noisy or wider distribution from the COSMIC-1 data. But both data reveal a slightly larger (more positive) $\Delta\alpha$ RIE in the SH than in the northern hemisphere (NH). The hemispheric difference is more pronounced during the day. The daytime low-latitude PDFs appear to have a longer tail at negative $\Delta\alpha$ values, which is consistent with the statistics from Spire data [Fig.7]. The Spire $\Delta\alpha$ PDFs have a width falling between COSMIC-1 and COSMIC-2, but show the similar hemispheric difference.

The $\Delta\alpha$ derived from the meteorological satellites (i.e., MetOp-B and FY3-E) [Figs.8-9], with a weaker signal in SNR (signal-to-noise ratio) and a higher orbital altitude, tends to have a noisier PDF than those from COSMIC-1/2 and Spire constellations. It remains unclear whether the noisier behavior is related to sampling/orbital parameters or to the RO receiver design and their observing environment on spacecraft. Both MetOp and FY3 employed a very different receiver design from those used by COSMIC and Spire. In addition, the GNSS-RO receivers from COSMIC and Spire are considered as the

primary payload on the spacecraft without much interference from other instruments on board. Interference and multipath issues can be a susceptibility problem for the multi-instrument spacecraft like MetOp and FY3, which may affect the RO receiver for making high-quality ionospheric measurements.





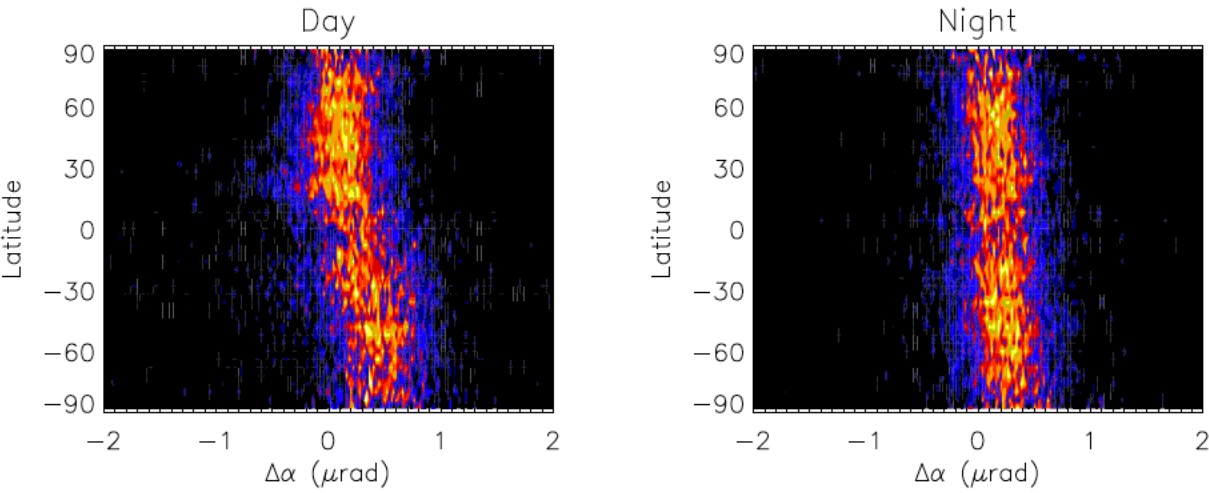


Fig.5 Latitude dependence of the probability density function (PDF) of COMSIC-1 Δα in μrad for day and night from Jan 2013. The PDF is normalized independently to its peak value at each 4° latitude bin. The PDF colors vary linearly between 0 (black) and 1 (yellow).

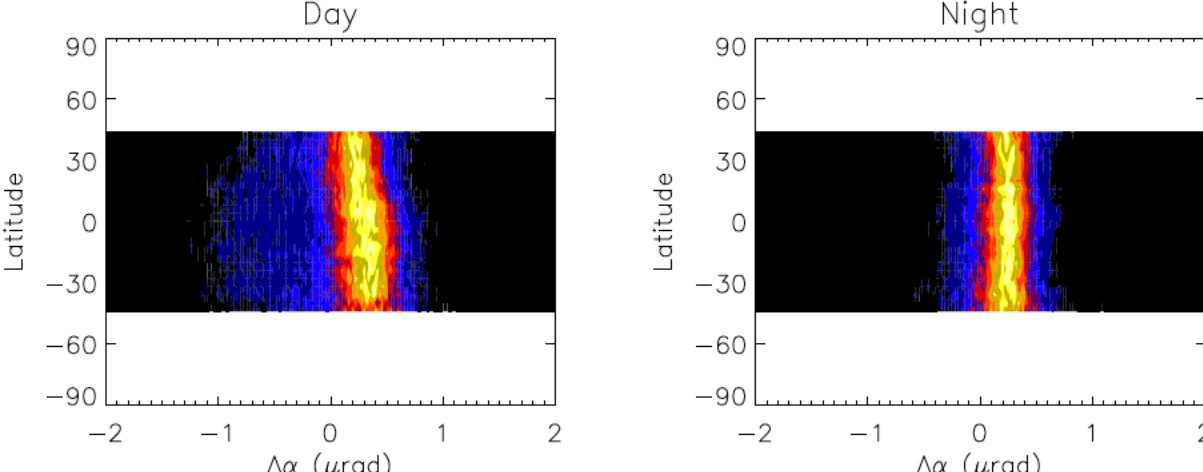

Fig.6. As in Fig.5 for COMSIC-2 from Jan 2022.





Fig.7. As in Fig.5 for Spire from Jan 2022.

Fig.8. As in Fig.5 for MetOp-B from Jan 2023. MetOp-C (not shown here) has very similar $\Delta\alpha$ PDFs to MetOp-B.





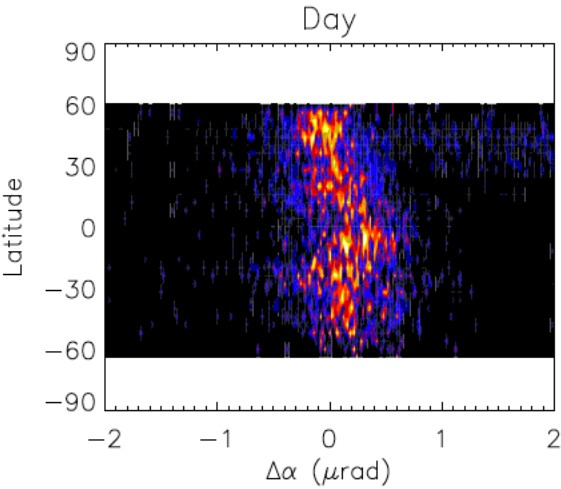
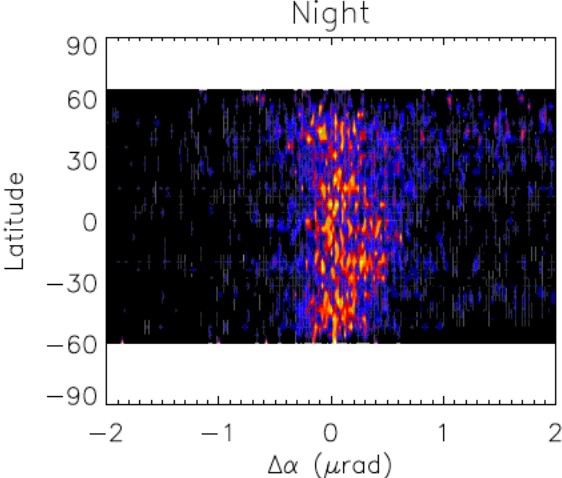


Fig.9 As in Fig.5 for FY3-E from Jan 2023.

## 3.2 Diurnal and Solar-Cycle Variations

The mean ($\mu$) and standard deviation ($\sigma$) of $\Delta\alpha$ from January and July 2013 show non-uniform distributions that vary with local solar time (LST), geographical location, season and solar activity [Figs. 10-11]. To illustrate the LST and

latitudinal variations, we used the COSMIC-1 measurements from January and July 2013 when solar activity was near its maximum. The monthly data are aggregated to 4° latitude and 2-h LST bins, using the quality screening criteria as shown in Table 1 with QC#4 (120 km) and QC#5. For the solar cycle variation, we used all COSMIC-1 data from 2006 to 2019 and averaged all-LST to produce a time series of monthly zonal mean. During most part of its mission (2007-2017), COSMIC-1 had a full diurnal sampling from its 6-satellite constellation on a precessing orbit. But in the later period, failure of some

satellites degraded the diurnal sampling and yield a slightly noisy result in the time series.

There are significant differences between the $\Delta\alpha$ morphologies derived from the $\phi_{ex}$-gradient method [Figs.10-11] and the $\kappa$ method [Fig.12]. The $\phi_{ex}$-gradient method reveals both negative and positive values in the $\Delta\alpha$ distributions, with mostly negative during the daytime and positive at night, whereas the $\kappa$-method always produces a negative RIE value from the product of a negative $\kappa$ value and $(\Delta\alpha_1 - \Delta\alpha_2)^2$, regardless of day and night. To further illustrate their differences, we

applied the same method (Eq.7) separately to the $\phi_{exL1}$ and $\phi_{exL2}$ data, and derived $\Delta\alpha_1 \equiv -d\phi_{exL1}/dh_t$ and $\Delta\alpha_2 \equiv -d\phi_{exL2}/dh_t$. Unlike the $(\Delta\alpha_1 - \Delta\alpha_2)^2$ distribution derived using an ionospheric electron density model [Angling et al., 2018], the $(\Delta\alpha_1 - \Delta\alpha_2)^2$ distributions and variations derived from the multi-year COSMIC-1 RO data [Fig.12] are more realistic, and they can be properly compared to the $\Delta\alpha$ morphology derived from the $\phi_{ex}$-gradient method.

Since the $(\Delta\alpha_1 - \Delta\alpha_2)^2$ distribution was introduced to provide a leading-order ($f^{-2}$) correction of the ionospheric

impact on the radio wave propagation, it is expected that the higher-order RIEs may be not captured by the $\kappa$ method. In other words, the distribution and variation of $\Delta\alpha_1 - \Delta\alpha_2$ difference cannot be fully characterized by $(\Delta\alpha_1 - \Delta\alpha_2)^2$ . For the





RIE impact on retrievals of temperature and humidity, $\Delta\alpha_1 - \Delta\alpha_2$ represents the persistent bias term in the inversion problem applied to the $\alpha$ data. As seen in Fig.10, not only two subtropical $\Delta\alpha$ peaks are negative during the daytime, but they also exhibit different mean values with a larger magnitude in the NH. This subtropical difference is much less

pronounced in the $(\Delta\alpha_1 - \Delta\alpha_2)^2$ distribution [Fig.12]. In addition, the summer-winter contrast (January vs July) is more pronounced in the $\Delta\alpha$ distribution than in the $(\Delta\alpha_1 - \Delta\alpha_2)^2$. The seasonal and hemispheric differences between the $\Delta\alpha$ and $(\Delta\alpha_1 - \Delta\alpha_2)^2$ morphologies are reflected in their solar cycle variations as well. The $\kappa$-method correction using $(\Delta\alpha_1 - \Delta\alpha_2)^2$ shows a variation symmetric about the equator [Danzer et al., 2020], similar to the distribution revealed in Fig.12c. Depending on how the RIE is derived, the application of two different RIE corrections as described above can likely impact

on the neutral atmospheric retrievals and assimilation of the $\alpha$ data.

Solar-cycle variations are more pronounced in the daytime $\Delta\alpha$ than in the nighttime [Fig.11], as expected the RIE associated with for the strong daytime photoionization in the ionosphere. From the $\phi_{ex}$-gradient method, the $\Delta\alpha$ time series shows a larger daytime negative RIE with a hemispheric asymmetry during the solar maximum years, but a larger nighttime positive RIE during the solar minimum years. At high latitudes, a summertime positive RIE appears to be a repeatable

phenomenon with slightly higher values during the night in the NH. There is an indication of weak solar-cycle variations in the daytime high-latitude $\Delta\alpha$. The solar cycle variations from COSMIC-1 are consistent with the MetOp-A/B/C observations (not shown), which have global coverage at two fixed LSTs. In summary, the solar-cycle variation of the RIE derived from $\Delta\alpha$ differs substantially from those from the $\kappa$-method based on $(\Delta\alpha_1 - \Delta\alpha_2)^2$. The time series of RIEs estimated by the $\kappa$-method exhibits little high-latitude variation with a similar solar cycle in both NH and SH subtropics during the day.

In addition to the mean ($\mu$) distribution of $\Delta\alpha$, a large variability of $\Delta\alpha$ may also play an important role in the RIE correction. If the underlying processes are not randomly and linearly varying with RIE, a spatial or temporal average would not eliminate RIE impacts on the neutral atmosphere retrievals. It is reasonable to assume ionospheric fast processes may impact atmospheric $\alpha$ in a nonlinear way, which could result in a residual mean that depends on its variance. To further evaluate the $\Delta\alpha$ variation and its dependence on the $\Delta\alpha$ standard deviation ($\sigma$), we compute the monthly maps for both

variables on a latitude x longitude (4°x8°) grid for every 2-h LST bin. Fig.10 shows the LST-and-latitude distributions of $\Delta\alpha$ $\mu$ and $\sigma$ from Jan and July 2013. Both variables exhibit a strong diurnal variation, which is latitude-dependent with a large $\sigma$ amplitude at mid-latitudes and during the nighttime. The correlations between the $\Delta\alpha$ $\mu$ and $\sigma$ distributions are complicated, but in both months the $\Delta\alpha$ $\sigma$ amplitude can be greater than the $\Delta\alpha$ mean, particularly in the nighttime. The season-dependence of the $\Delta\alpha$ $\sigma$ diurnal variation shows a larger amplitude at mid-latitudes in the winter daytime/evening and in the

summer nighttime.

**Fig.10.** Latitude (4°-bin) and solar local time (2h-bin) dependence of COSMIC-1 RIE $\Delta\alpha$ mean ($\mu$) and standard deviation ($\sigma$) from January and July 2013.

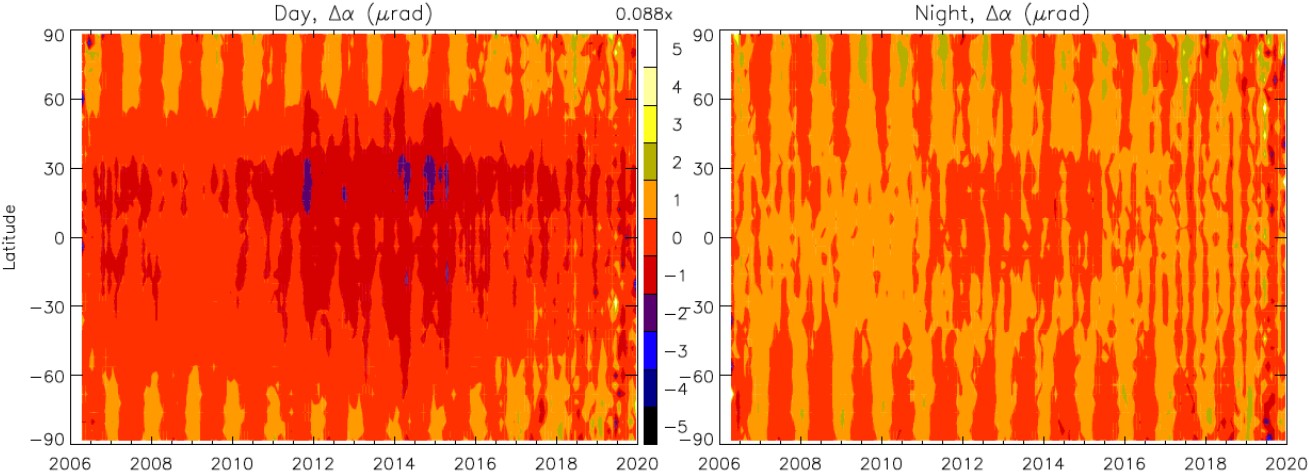

**Fig.11**. Long-term variations of the daytime and nighttime mean $\Delta\alpha$ from COSMIC-1 as a function of latitude during 2006-2020.

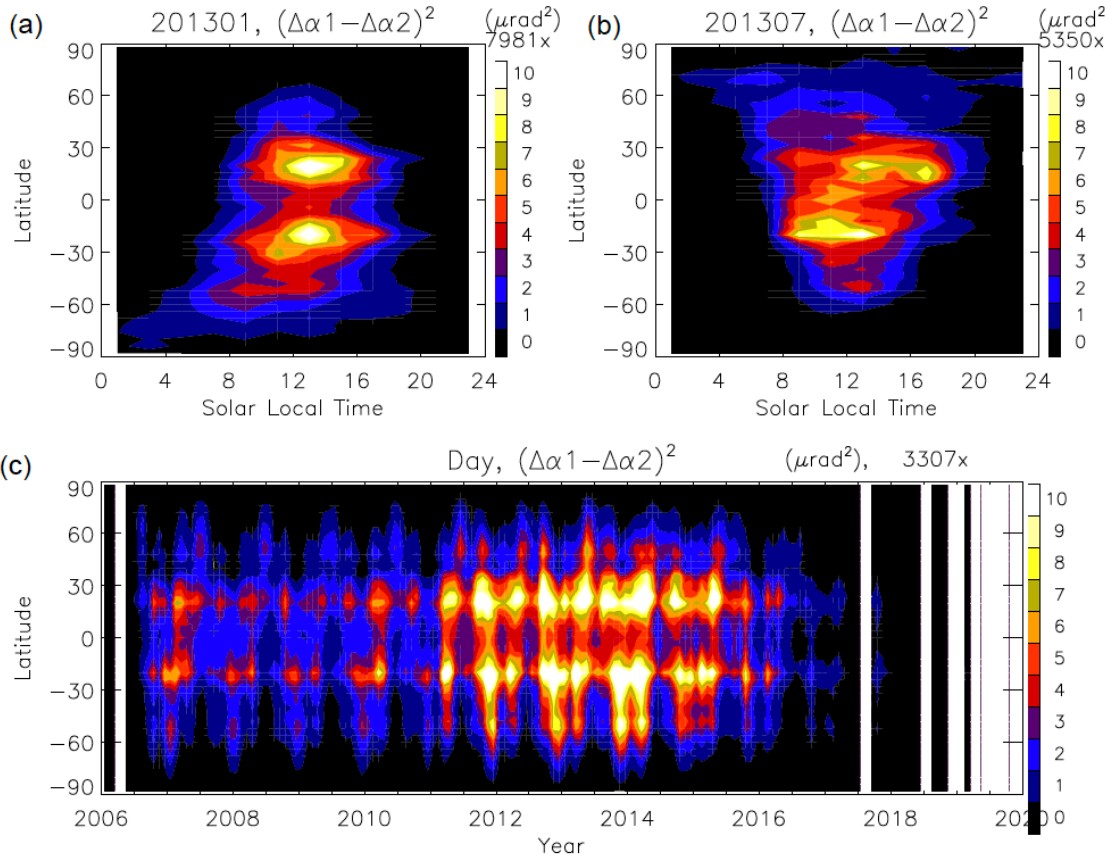



**Fig.12.** The $(\Delta\alpha_1 - \Delta\alpha_2)^2$ distribution and variations derived from the COSMIC-1 data using $\Delta\alpha_1 \equiv -d\phi_{exL1}/dh_t$ and $\Delta\alpha_2 \equiv -d\phi_{exL2}/dh_t$. The $(\Delta\alpha_1 - \Delta\alpha_2)^2$ distribution here can be compared to the RIE derived from the $\kappa$-method in Fig.10, by multiplying a $\kappa$ value between -10 and -15 rad$^{-1}$ for 60 km.

### 3.3 Longitudinal Variations and Dependence on Geomagnetic Field

As well recognized [Forbes et al., 2006; Immel et al., 2006], the ionospheric variability is also driven by atmospheric waves originated in the lower atmosphere and the geomagnetic forcing from above. As a result, the underlying ionospheric processes that cause the RIE are likely dependent on the geomagnetic activity as well as on the longitudinal and seasonal variability of waves. Thus, it is important to quantify the $\Delta\alpha$ distribution and its variations in the geographical and geomagnetic coordinate systems. The monthly averaged geographical $\Delta\alpha$ maps derived from COSMIC-1 data for January and July 2013 show large nighttime longitudinal variations of $\Delta\alpha$ [Fig.13], which may be induced by the atmospheric planetary waves forced in the lower atmosphere and mesosphere. There is also an indication that the daytime values of $\Delta\alpha$ vary slightly with the geomagnetic field at low and middle latitudes in both seasons. At high latitudes, however, there is no indication of noticeable $\Delta\alpha$ variations and their links to the auroral perturbations in the polar caps. The dependence of RIE on the geomagnetic field variation is expected for path differences between the L1 and L2 propagation in the ionosphere [Hoque and Jakowski, 2007], and the amplitude of RIEs can be both positive and negative [Vergados and Pagiatakis, 2011].

It is interesting to remark that the nighttime $\Delta\alpha$ distributions from the summer and winter bear a strong resemblance to those derived for the sporadic E (Es) climatology [Wu et al., 2005; Arras et al., 2009]. Previous studies with global GNSS-RO observations have showed that the occurrence of Es tends to peak at 90-110 km altitudes near mid-latitudes during the summer months. The occurrence of Es-layers is strongly modulated by the solar diurnal and semidiurnal migrating tides. Syndergaard [2000] discussed the Es-induced morphology of RIE showing a frequent influence of Es below the occurrence altitude. This study also suggested the RIE correction scheme for the Es-induced errors.

However, it remains unclear to what extent Es may contribute to the RIE amplitude and variability. Although the $\phi_{ex}$-gradient method attempts to minimize the Es impacts on the RIE calculation [Fig.2], the RIE maps from Fig.13 seem to indicate that Es may play a significant role in the nighttime RIE variation. Because the E- and F-region ionospheric variabilities are driven by different processes, their contributions to the RIE would appear as different morphologies in terms of latitudinal, longitudinal and local time distributions. As elucidated by Syndergaard and Kirchengast [2022], path differences between the L1 and L2 propagation in a 3D structured ionosphere are the major cause of various RIEs, which can vary with the geomagnetic field and the spatial distribution and gradient of electron density. Depending on the relative importance of these contributors, the RIE corrections and impacts on the neutral atmospheric measurements are likely to differ from each other.





Fig.13. Geographical maps of the $\Delta\alpha$ derived from COSMIC-1 $-d\phi_{exL1}/dh_t$ measurements for January and July 2013; the white lines display positions of the geomagnetic equator.

# 4      RIE Impacts on Data Assimilation (DA)

## 4.1      Goddard Earth Observing System for Instrument Teams (GEOS-IT)

To quantify potential impacts of the observed $\Delta\alpha$ morphology on the neutral atmosphere, we conducted several data assimilation (DA) experiments using the GEOS-IT DA configuration of NASA GMAO/GSFC https://gmao.gsfc.nasa.gov/GMAO_products/GEOS-IT). GEOS-IT retains many characteristics of GEOS Forward Processing System, including the spatial resolution (~50 km) and the use of a three-dimensional variational (3D-Var) DA algorithm. It allows the instrument teams to benefit from many model enhancements of GEOS, leading to more realistic



representations of moisture, temperature, land surface, etc., and analysis changes that introduce the most modern new satellite observations into the system. The GEOS-IT configuration employed in this study is based on the GEOS-5.27 DA

system. GEOS assimilates GNSS-RO $\alpha$ observations with a 6-hour update cycle [McCarty et al., 2016; Gelaro et al, 2017], using the RO forward operator as in the operational NCEP (National Centers for Environmental Prediction) bending angle method (NBAM) [Cucurull et al., 2013]. As shown in Cucurull et al. [2014], the GNSS-RO observations have both direct and indirect impacts on the quality and skills of analyses and forecasts of NWP systems. The assimilation of RO in an operational NWP system of NOAA results in a more accurate bias correction for infrared and microwave radiance

measurements, leading to more effective use of satellite radiances with larger number of radiance data that pass quality control. This indirect impact has made the GNSS RO observation more valuable even is number of profiles is relatively small compared the radiance measurements. Because the GNSS-RO technique is essentially traced to the SI length unit, it depends little on radiometric calibration as with those microwave and infrared sounders and can serve as an anchor for the radiance bias correction schemes for the nadir-viewing sensors.

The direct impact of RO data comes from the $\alpha$ sensitivity to temperature perturbations in the upper troposphere and stratosphere. The RO measurements constrain the tropospheric water vapor and stratospheric temperatures below 40 km with the global and full local time coverage. The ROPP package employed to produce the $\alpha$ data and their errors from the $\phi_{ex}$ data [Healy et al., 2007; Cucurull et al., 2013, 2014; Zhang et al., 2022]. The GEOS DA algorithm for the RO observations assumes the zero bias in $\alpha$ data and does not consider the RIE correction schemes. As remarked in previous DA

studies above ~30-40 km [Healy et al., 2007], the RIE correction schemes for the $\alpha$ data analysis become important, and it needs to be introduced to address the systematic $\alpha$ errors associated with the persistent influence of the ionospheric processes. To minimize the RIE impact, the covariance matrix in the DA system is configured to weigh less of the $\alpha$ data from altitudes above 40 km. However, RIE impacts are still evident in the resulting analysis data [Danzer et al., 2013]. In the next section we introduce the $\phi_{ex}$-based RIE correction scheme as described in Section 3 and carry out a set of DA

experiments to quantify the impact of RO data with and without the RIE on the GEOS-IT analyses.

## 4.2    DA Experiments

To evaluate the RIE impacts inferred from the $\phi_{ex}$-gradient method (Section 2.2) we performed the GEOS-IT experiments for the Dec-Jan of 2016/2017. During the selected period, about 2000 $\alpha$ profiles per day were analyzed from multiple GNSS-RO missions (i.e., GRACE, Metop-A, and COSMIC-1). The horizontal resolution of GEOS-IT analyses and

forecasts is ~50 km, and results were archived at 72 model layers from the surface to ~1Pa (~80 km). We conducted four different experiments: (1) control (CTL) experiment that assimilate all RO observations assuming no RIE bias; (2), denial (NoGPS) experiment by excluding all RO data; (3) constant bias (BiasM2) experiment by adding a large constant RIE ($\Delta\alpha$ = -2 µrad) to all RO data; and (4) realistic bias experiment (BiasLST) by incorporating the latitude-LST dependent RIE in $\alpha$ that is similar to the COSMIC-1 observation in Fig.14.





The month of December 2016 was used to spin up the GEOS-IT analysis-forecast system, and the RO data were injected starting from January 1 2017 for the RIE-impact experiments. The objective of these experiments is to quantify the potential impacts of the RIE bias from assimilating the $\alpha$ data with the recent upgrades of the GEOS-IT system (model and DA algorithms). We compared the Jan 1-10 GEOS-IT analyses as differences relative to the CTL experiment displaying differences in the zonal mean temperatures and highlighting potential impacts of the diurnally varying RIE on the DA in the

middle and upper atmosphere. The importance of RIE impacts depends on the RIE amplitude relative to atmospheric variability at each altitude and how much the GNSS-RO measurements are weighted in the DA system. The latter is defined by the specification of error statistics for the $\alpha$ data. We used the identical GEOS-IT $\alpha$ error covariance matrix in all experiments for the RO data assimilation. The $\alpha$ error covariance matrix was designed to reduce the weighting on the RO $\alpha$ data at higher altitudes so that their large relative errors there does not impact the lower atmosphere. Nevertheless, the DA

results from the $\alpha$ data with a RIE can still have a significant impact on the middle atmosphere.

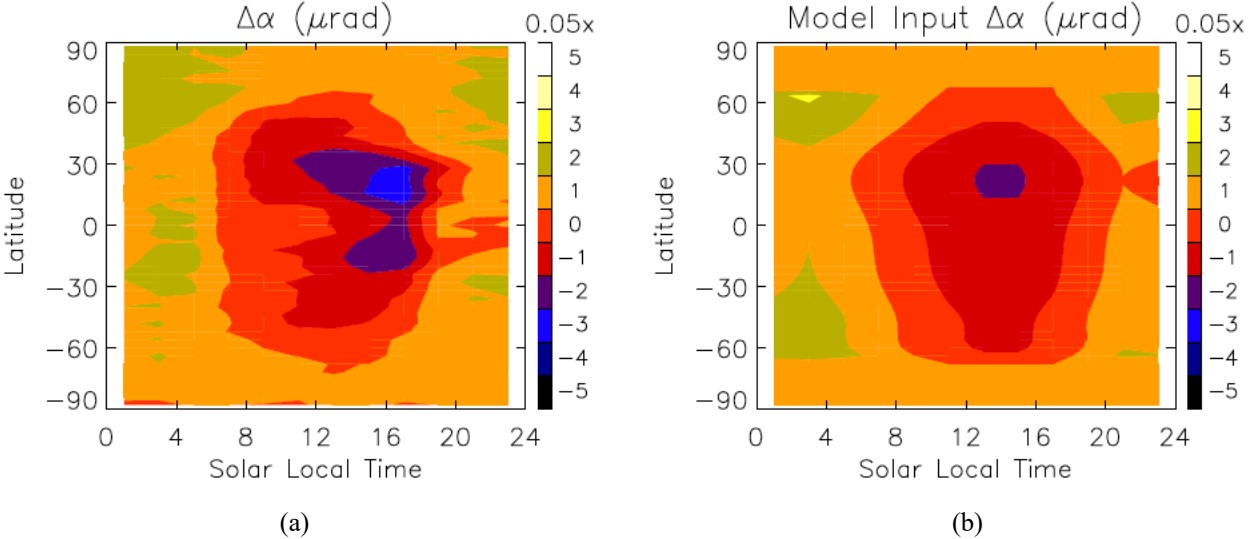

(a)                              (b)

Fig.14. (a) Annual mean $\Delta\alpha$ variations derived from 2013 COSMIC-1 data; (b) Parameterized $\Delta\alpha$ variations used as the RIE
input for the "BiasLST" experiment.

### 4.3     Temperature

        The RIE impacts on the DA results tend to grow with time and height in amplitude, despite the reduced sensitivity to the RO $\alpha$ measurement in the upper atmosphere. We chose the 5-7 day period after the GPS measurements are injected with a bias or denial, to characterize an accumulated impact from the spatiotemporal growth of the DA system. During the

initial growth period (0-5 days), the DA system continues to inject the biased/denied RO data and the rate of growth between CTR and other perturbed runs appears to be large. The growth rate of differences slows down on days 5-7, which allows a better characterization of the RIE impacts.





The 3-day (Jan 5-7) averaged temperature differences between the CTL and three sensitivity experiments (NoGPS, BiasM2, and BiasLST) show significant RIE impacts on the GEOS-IT analyses (Figs 15-17). The zonal mean differences

(Fig. 15) display the most prominent impacts at high latitudes in the upper GEOS domain ( above 10 hPa, ~30km) Compared to the NoGPS impacts (Fig.15a), the GNSS-RO data with a large (2 μrad) global bias (Fig.15b) illustrate that the unrealistic specification of the RO bias in the DA can initiate significant deviations between the GEOS-IT analyses (a 3-30 K deviations in the upper atmosphere and 0.3-0.6 K in the troposphere). The large biases in the upper atmosphere suggest that the DA system still has a significant weight on the GNSS-RO measurements at these altitudes and is biased toward the

error-prone RO observations. In the BiasLST case where the RIE bias amplitudes are more realistic with ~0.05 μrad during the daytime, the impact of the RIE bias correction reduces to ~1-3 K in the upper stratosphere. The magnitudes of temperature error introduced by realistic RIE specification (Fig. 14) are comparable to the results reported by Danzer et al. [2020] with the κ-method correction.

In the lower troposphere the GNSS-RO contribution to the DA comes primarily from an indirect impact in which

the RO data help to correct the radiance bias between microwave and infrared sounders and allow assimilation of more microwave radiances [Healy et al., 2005; Cucurull et al., 2014]. The CTL-NoGPS differences (Fig 15b) show that the assimilation of GNSS-RO data help to improve the temperature in the polar regions by ~0.3-0.9 K. However, if the GNSS-RO data have a large bias (BiasM2, Fig 15d), the assimilation with the RO data would lead to a large (~0.6 K) temperature deviation relative to the CTL run. If the RIE bias of GNSS-RO data is small, as in the BiasLST experiment, the contributions

of RIE are insignificant in the lower troposphere.

It is interesting to observe that the large GNSS-RO impacts are mostly located in the high latitude polar regions, which appears to be the case for the denial experiment (NoGPS) as well as for the experiments with the bias specification. These results are consistent with some of the earlier studies with RO-denial experiments, showing larger impacts at higher latitudes [Bonavita, 2014; Cucurull and Anthes, 2015]. Again, the loss of GNSS-RO observations and the correction of

microwave radiance biases are highly correlated. Both direct and indirect impacts from the GNSS-RO observations may act collectively in the large temperature differences seen in the polar region [Healy et al., 2005].

In the upper atmosphere where the solar migrating tides dominate the diurnal variation in temperature and winds, it is imperative to assess the RIE impact as a function of local time for its relative importance. Figs.16-17 show the local time variations of temperature at 1 and 0.1 hPa where the GNSS-RO RIE errors can be very large, compared to the mean $\alpha$.

Interestingly, the local time variations from experiments NoGPS and BiasLST look very similar to that from the Control experiment, despite large differences in the zonal mean temperatures. The experiment with a very large (BiasM2) appears to damp the diurnal amplitude by 30-50%. Otherwise, the experiments with NoGPS and BiasLST do not seem to significantly change the tidal phase in LST because the dominant migrating tidal modes are strongly locked in phase to the solar forcings in the troposphere and stratosphere.








Fig.15. The zonal mean temperature differences between controlled and perturbed experiments. A different color scale is used for the lower atmosphere, to highlight small values in this region.




Fig.16. Diurnal temperature variations from the controlled and perturbed experiments at 1 hPa. The temperature perturbations from its zonal mean in K are contoured in color.








Fig.17. As in Fig.16 but for 0.1 hPa.

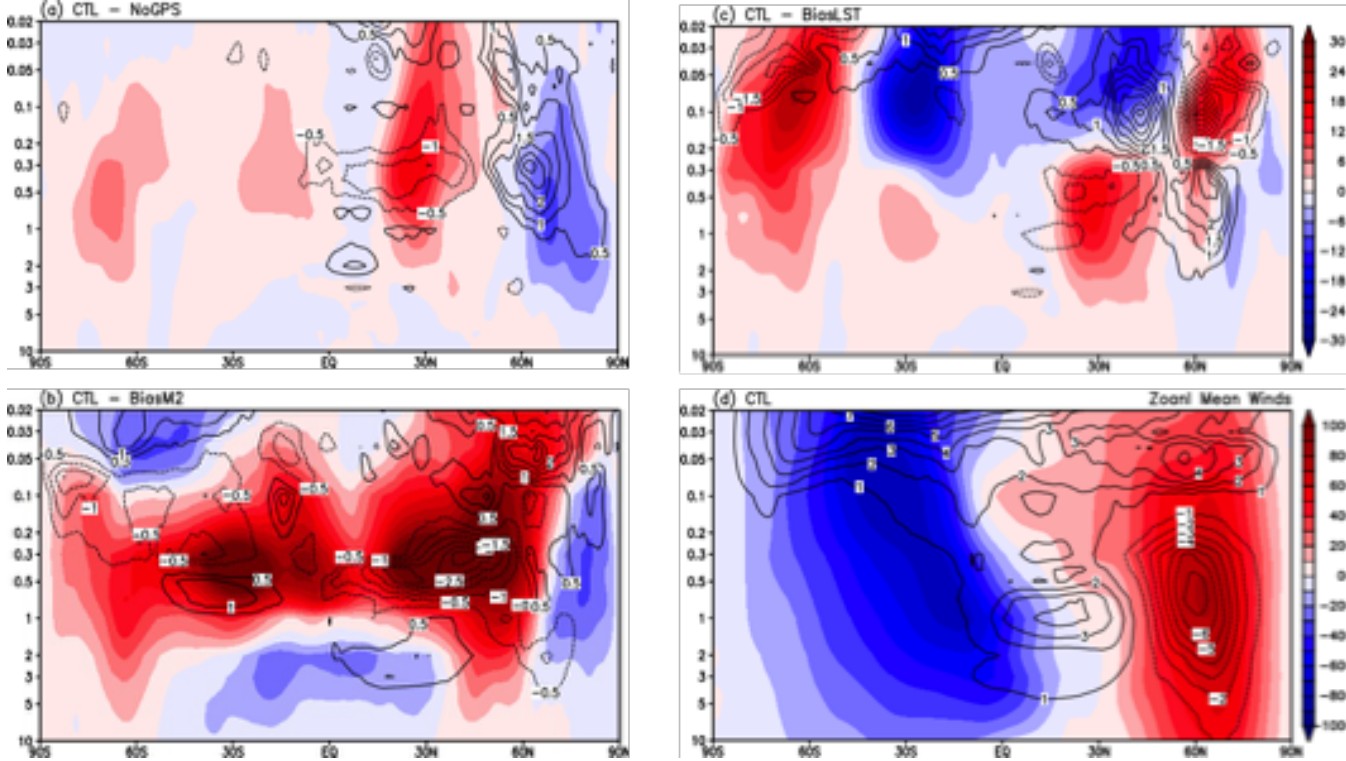

Fig.18. Impacts of the RIEs on upper-atmospheric winds at 10-0.02 hPa from the DA experiments: (a) CTL-NoGPS, (b)
CTL – BiasM2, and (c) CTL – BiasLST, sharing the same color scale in panel (c). The mean wind from CTL is displayed in
(d). All panels have the zonal winds contoured in color with unit of m/s, superimposed by the meridional winds contoured by
lines.

Despite a relaxed value used in the DA covariance matrix in assimilating GNSS-RO $\alpha$ from higher altitudes, the
measurement errors such as RIE still have a significant impact on the assimilated winds and temperatures at pressures < 5
hPa. For example, in Fig.18 the CTL – NoGPS difference suggests that the GNSS-RO data made an equatorward shift of the
polar vortex near 0.2 hPa in the northern hemisphere winter. In the presence of a RIE like BiasLST, this shift becomes larger
at pressures > 0.2 hPa but in the poleward direction at <0.2 hPa. In the southern hemisphere, the BiasLST makes the zonal
winds shifted significantly equatorward. Because the reanalysis data have been widely used at these pressure levels (1 – 0.1
hPa), it would require a special attention to the RIE impact during the period when GNSS-RO data are assimilated.






## 5 Discussions

### 5.1 Negative and Positive RIE Values

Contributing factors to the RIE are complex because higher-order effects on the refractive index can come from various sources during the radio wave propagation through a structured and anisotropic ionosphere [e.g., Davies, 1965; Kindervatter and Teixeira, 2022]. As modelled by Brunner and Gu [1991], the higher order terms from the series expansion of the refractive index include the first-order effect from the path difference between the L1 and L2 frequencies and the second-order effect from the geomagnetic field. While the first-order RIEs are mostly negative, the second-order RIEs can be both

positive and negative. To quantify the RIE for ground-based receivers, Hoque and Jakowski [2007] developed a correction algorithm that can be applied to real-time GNSS applications and reduce the higher-order phase errors.

In the GNSS-RO applications, studies have also found that the first-order RIEs are mostly negative [Liu et al., 2013] and the second-order could have both positive and negative values in $\alpha$ and refractive bias depending on the viewing geometry with respect to the geomagnetic field [Vergados and Pagiatakis, 2011; Qu et al., 2015; Li et al., 2020]. These

model simulations all suggested a relatively small RIE value that is typically less than 0.1 μrad, generally smaller than what was observed with the $\phi_{ex}$-gradient method introduced in Sections 2 and 3. Nevertheless, the positive and negative $\Delta\alpha$ values seen in Figs. 5-9 suggest that the first and second-order RIE effects are equally important. Examining the mean deviation between CHAMP and COSMIC-1 $\alpha$ and a climatology model at 60–80 km altitudes, Li et al. [2020] also reported positive and negative values of RIE, showing larger negative RIEs in the daytime and relatively smaller positive RIEs at

night as seen in Fig.10. Note that this study did not use any climatology model for $\alpha$, and the $\phi_{ex}$-gradient method is self-sufficient with an empirical linear fitting to each individual profile. In summary, compared to the idealized model simulations, the large variability in the observed positive and negative $\Delta\alpha$ reflects the complex nature of RIEs in the dynamical ionosphere.

### 5.2 Impacts of Es and RO Top Height

An Es layer can produce large RIEs near the layer heights with a long tail extended to lower tangent heights in the GNSS-RO profile [Syndergaard, 2000; Syndergaard and Kirchengast 2022]. If the RO measurements stop at or below the Es layer, the $\phi_{ex}$-gradient method can be significantly affected by the Es tail and lead to an overestimated RIE. In some GNSS-RO operations (e.g., MetOp), the top of RO data acquisition is often capped at ~85 km. To quantify effects from this low RO top height, we took advantage of the experimental MetOp-A data on 2020d161-2020d254 (June 9-September 10, 2020)

when the high-rate (100 Hz) GNSS-RO acquisition reached up to ~290 km. This data set allows us to quantify the RO top impact from different truncation heights.





In this comparative analysis, the $-d\phi_{ex}/dh_t$ RIE algorithm was applied to the same experimental MetOp-A data from June 9-September 10, 2020, but from different RO truncation tops: 90, 120 and 170 km. The 90-km truncation represents some missions before 2020 such as MetOp and SACC when the high-rate RO data acquisition stopped at ~85 km.

The 120-km truncation height have been adopted by several missions (e.g., COSMIC-1, TSX, FY3), whereas the 170-km top operation corresponds to some of the recent commercial GNSS-RO constellations (e.g., Spire and PlanetiQ). More detailed information on the RO sampling parameters of the past and current GNSS-RO missions can be found in Appendix A.

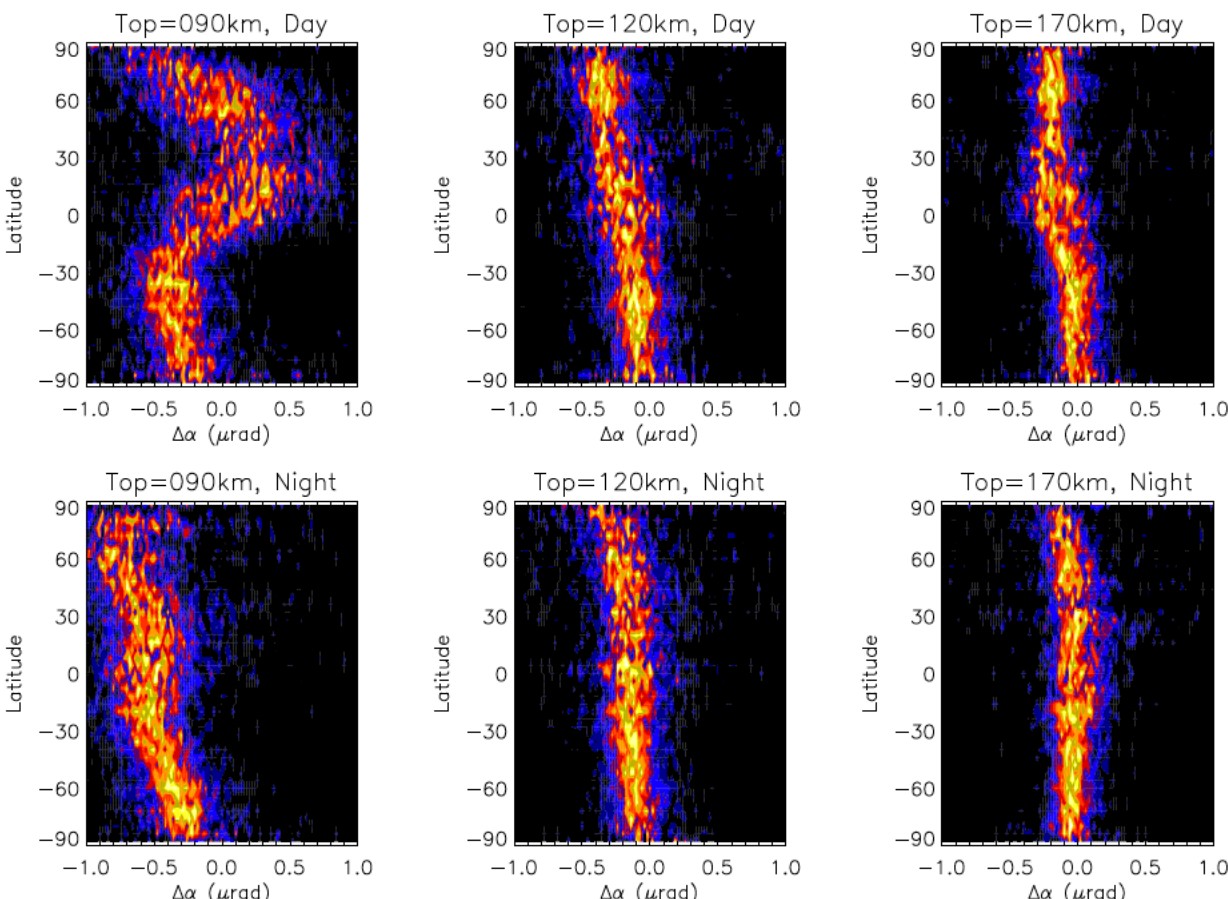

Fig.19. Probability distribution function (PDF) of the RIE derived from the $-d\phi_{ex}/dh_t$ algorithm to compare impacts of the

RO top height truncation. The experimental data from the MetOp-A special scan during D161-D254 in 2020 were used in the analysis and the RIE statistics from day (top panels) and night (bottom panels) are reported separately. The RIE results from three RO cutoff top heights, 90, 120, and 170 km, are selected for comparisons.

Fig.19 shows that it is imperative to have the RO sounding top reaching at least 120 km, because of the inconsistent RIE results from the 90-km truncation calculation. The RIEs derived from the 120-km and 170-km truncations have similar

statistics for both day and night, suggesting that the 120-km truncation could overcome the potential Es influence in the RIE





calculation. As shown in Fig.2c, the $\phi_{ex}$-gradient method would suffer from the Es tail below the Es layers. By providing a few RO measurements above the Es layer, it would help substantially to constrain $-d\phi_{ex}/dh_t$ for the RIE calculation. Therefore, the inconsistent statistics from the 90-km truncation, compared to those derived from a higher RO top, can be explained primarily as the Es impacts. If one plans to use the $\phi_{ex}$-gradient method to estimate and correct the RIE, the RO

top height as summarized in Appendix A becomes an important parameter to know for the current and past missions.

## 5.3    Other Methods

### 5.3.1 Deviation from the Exponential

RIEs and their effects can be evaluated with other methods or comparative analyses by including an $\alpha$ bias against ground-based radar observations [Danzer et al., 2013] and $\alpha$ climatologies [Li et al., 2020]. Here, we introduce a method that

uses the height variation of the excess phase $\phi_{ex}$ profile in the upper stratosphere, to compare it with the expected exponential lapse rate. Since $\alpha$ is related to the $d\phi_{ex}/dh_t$ [Eq.6], the $\phi_{ex}$ profile contains information on the RIEs seen in $\alpha$. It is shown in Melbourne et al. [1994] and Wu et al. [2022] that the iono-free $\phi_{ex}$ profile is proportional to

$$\phi_{ex}(h_t) \approx 1.8 \times 10^{-5} \cdot N(h_t)\sqrt{H \cdot R_e} \tag{8}$$

where $N$ is atmospheric refractivity, $H$ is the atmospheric scale height, and $R_e$ is Earth radius. Because the refractivity is

proportional to air density, it tends to decrease exponentially with height, $N(h_t) = N_0 e^{-(h_t-h_0)/H}$, from a reference $N_0$ at height $h_0$. Hence, a departure from the exponential height dependence as described by Eq.(8) may be used to evaluate an RIE in the $\phi_{ex}$ profile.

Several lapse rate comparisons are shown in Fig.20 where the $\phi_{ex}$ profile differences are highlighted in percentage from an exponential fit at heights above 40 km. The exponential model uses the data at 40-45 km for the fitting and

extrapolate the model to the heights above and below. The model assumes a constant scale height at these altitudes, which may cause some errors if atmospheric temperature varies by 10-20%. However, any large percentage departures would raise a concern and might indicate a RIE in the $\phi_{ex}$ profile. Since Eq.(5) cannot completely remove the ionospheric contribution, the departure from the exponential can be used to better the nature of these RIEs.  As discussed above, ionospheric multipaths and horizontal/vertical gradients might play a significant role in the RIE of the $\phi_{ex}$ measurements.

Fig.20a is a typical case where the exponential model fits well to the $\phi_{ex}$ profile up to ~50 km but exhibits negative biases (observation minus model) at higher altitudes. These biases can be as high as 40-50% at 50-60 km, much greater than 10-20% typically seen at lower altitudes. It is possible that the low biases were caused by a colder atmosphere at higher altitudes compared to the temperature at 40-45 km, because the scale height $H$ is proportional to temperature and $\phi_{ex}$ is proportional to $\sqrt{H} \cdot e^{-(h_t-h_0)/H}$ in Eq.8. For example, to explain a 40% decrease in $\phi_{ex}$, it would require that temperature

drops from 270 K at 45 km to 210 K at 55 km, which might be feasible from a very strong gravity wave.

Fig.20b illustrates a likely ionospheric effect where a sharp thin layer at ~53 km creates a large disruption in terms of exponential dependence of $\phi_{ex}$ with height. Not only the height dependence of $\phi_{ex}$ does not obey the normal exponential





decrease as expected from the neutral atmosphere, but also the departure from the exponential above the layer is severely lower by more than 50%. It remains unclear why the layer disrupts more at heights above than below. In other words, if such

behaviour is representative for the ionospheric effects of thin layers (e.g., Es), their RIEs on the neutral atmospheric measurements might be small.

Fig.20c shows very different exponential dependences between the atmosphere below 40 km and above. To explain the 70% difference using Eq.8, one would have to assume a temperature difference of 120 K between 25 km and 45 km. On the other hand, if a reference from the lower altitudes (e.g., 30 km) were used, it could help to reduce the biases in the lower

atmosphere but would induce a much larger (> 100%) bias at higher altitudes. In summary, this is a case likely caused by a RIE that can a large departure of the $\phi_{ex}$ measurement from the expected exponential dependence with height. The biases above and below 40 km are too large to be explained by a normal atmosphere, but a RIE in the $\phi_{ex}$ measurement can certainly induce a large error like this.

Fig.20d displays a wave-like oscillation above 40 km in the bias from the exponential dependence. It perhaps

reveals a real atmospheric gravity wave in the $\phi_{ex}$ measurement up to 60 km. If no RIE had impacted this profile, it would imply that the RO technique could provide good sensitivity to atmospheric temperature up to 60 km. Because RIEs can produce a local impact at a narrow height range like Es as well as an extended impact from the multipath propagation through the F-region, it remains a great challenge to distinguish between good cases like Fig.20d and RIE-impacted case like Fig.20b.

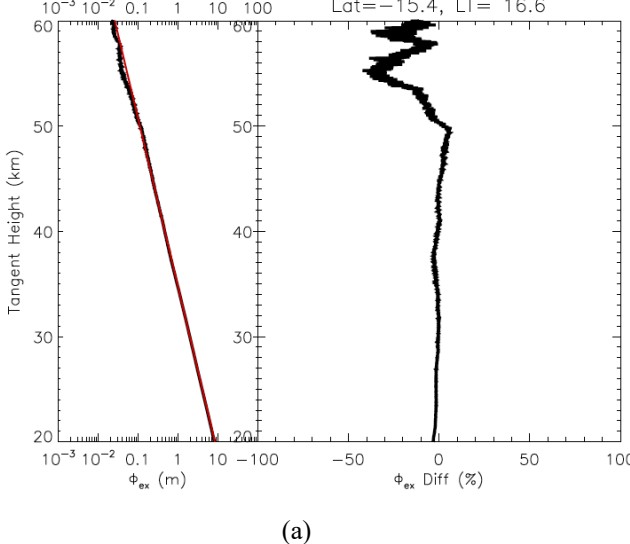

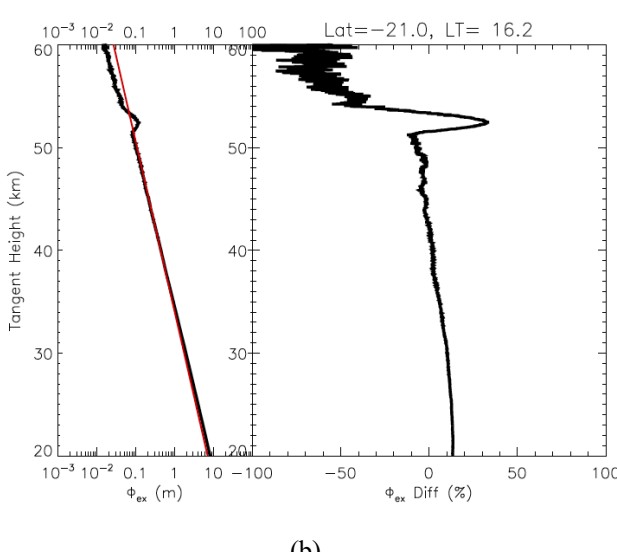


(a)                                             (b)


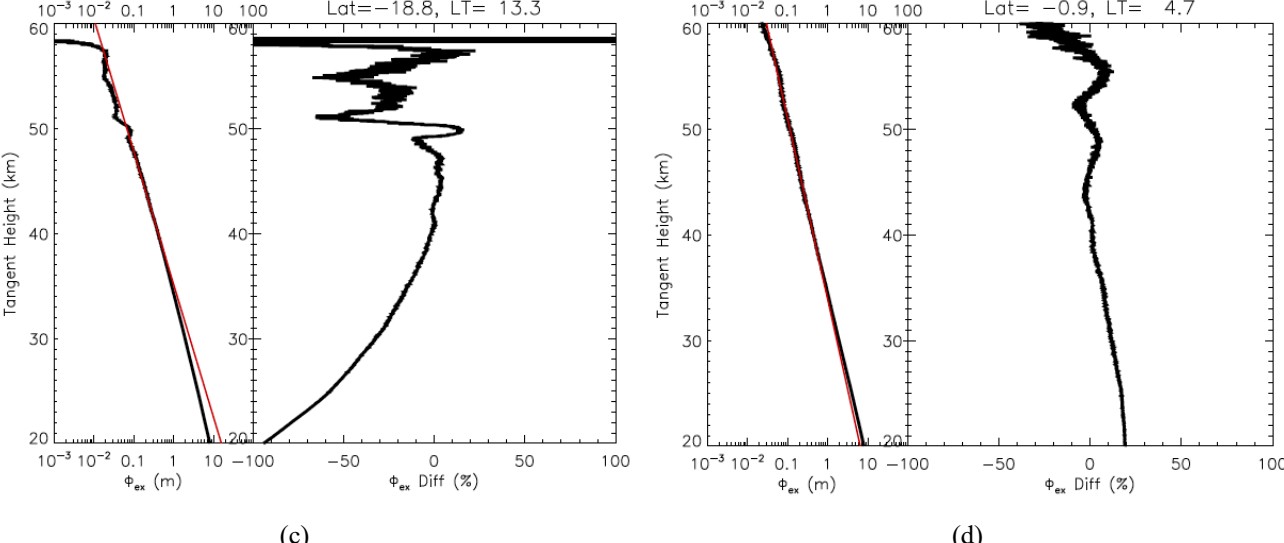

(c)                       (d)

Fig.20. Examples of the departure of excess phase ($\phi_{ex}$) measurements from an exponential fit (red line) at 40-45 km heights. The percentage difference between the observed and modelled $\phi_{ex}$ profile is displayed on the right as a function of tangent height. Selected cases are: (a) Typical negative departure from the exponential fit at higher altitudes; (b) Departure from the exponential function due to a layered structure near 52-53 km; (c) Largely different exponential dependence at heights above and below 40 km; and (d) Wave-like oscillations at high altitudes. These cases were extracted from COSMIC-2 observations on January 1, 2022.

### 5.3.2 Temperature Variances

Variability of the GNSS-RO temperature profiles can also be used to infer potential RIE impacts, because additional fluctuations induced by RIEs can be as large as the RIE-induced bias. As shown in Fig.10, the standard deviation of RIE $\Delta\alpha$ are often greater the mean, suggesting that these RIE $\Delta\alpha$ variations might have propagated to the atmospheric temperature retrieval. Because ionospheric variability has a wide range of spatiotemporal scales (e.g., scintillation, Es), it is possible that small-scale RIE $\Delta\alpha$ variations may be not completely removed and result in artificial wave-like oscillations in the retrieved temperature profiles.

To evaluate RIE impacts on RO retrieval variability, we analyzed the entire MetOp-A/B/C record of temperature data produced by the Radio Occultation Meteorology Satellite Application Facility (ROM SAF) to extract small-scale variances using the algorithm developed previously for Es and gravity wave (GW) studies [Wu et al., 2005; Wu 2006]. This algorithm is similar to those used in other studies [e.g., Tsuda et al., 2000; Schmidt et al., 2008], except that we derived GW variances using a bandpass filter for a more careful treatment on the measurement noise. In the data analysis with this algorithm, each RO temperature profile $T(z)$ is first processed with a running-window smoothing (truncation length $\Delta z$) to obtain a background ($\overline{T(z)}$). The difference $T(z) - \overline{T(z)}$ is used to compute the variance for this truncation $\Delta z$. The





minimum truncation length is defined by the 3-point length in the temperature retrieval, which is ~100 m in the upper troposphere and stratosphere and can be used to estimate the measurement noise on a profile-by-profile. The bandpass-filtered variance is defined as the variance difference between the $\Delta z$ and the 3-point truncation. The bandpass-filter method has been successfully applied to other satellite measurements for GW studies [Wu and Eckermann, 2008; Gong et al., 2012].


Fig.21. Time-height variation of the monthly small-scale variances derived from Metop-A/B/C RO dry temperature retrievals at 72.5°S.


Fig.21 shows a time series of monthly MetOp temperature variances derived with the bandpass-filter method for truncations 1 km and 2 km at 72.5°S. A significant solar-cycle variation is evident in the temperature variances, particularly in the upper stratosphere, which suggest that RIE impacts might have an amplitude of 0.03 K$^2$ and 0.3 K$^2$ in the 1-km and 2-km variances at ~40 km altitude. The similar solar-cycle variation exists in the northern high latitudes (not shown) with a



comparable amplitude, but the solar-cycle dependence becomes less pronounced at low latitudes. It is expected that the RIE impacts would be greater at high latitudes, as revealed in the DA impact study (section 4.3). Although the solar-cycle-dependent temperature variations do not provide a direct indication on whether the RO temperature is artificially biased by a solar cycle influence, the earlier studies have found such bias evidence [Danzer et al., 2013; Li et al., 2020].

## 6      Conclusions

In this study we developed a novel empirical algorithm for estimating the residual ionospheric error (RIE) in the RO bending angle ($\alpha$) data. The method, called the $\phi_{ex}$-gradient method, is self-sufficient and based on the vertical derivative of RO excess phase ($\phi_{ex}$) measurements. A linear fit was applied to the $\phi_{ex}$ data at heights above 65 km to determines the RIE ($\Delta\alpha = -d\phi_{ex}/dh_t$) in each individual RO profile without relying on any auxiliary data or model sources. The derived RIE is extrapolated to the RO measurements at the lower heights by assuming that $\Delta\alpha$ has the same impact on the entire $\alpha$

profile.

   Although the derived RIE ($\Delta\alpha$) climatology is dominated by positive values, the $\phi_{ex}$-gradient method also produces negative values for both day and night. This is fundamentally different from the $\kappa$-method that only produces the positive RIEs. Diurnal variations of the RIE $\Delta\alpha$ are latitude-and-LST dependent with larger negative amplitudes (up to -3 µrad) in the daytime tropics and subtropics. The standard deviations of RIE $\Delta\alpha$ can be greater than their mean values in the

climatology averaged by LST and latitude. Significant solar-cycle and longitudinal variations are found in the RIE $\Delta\alpha$ derived from observations. The dependence of RIE on geomagnetic field is evident but relatively weak, compared to the diurnal and geographical variabilities.

   RIE impacts on data assimilation (DA) were evaluated with the several experiments using the NASA GMAO Goddard Earth Observing System (GEOS) that assimilate GNSS RO $\alpha$ data for MERRA-2. A LST-latitude varying $\Delta\alpha$ bias

that approximates the COSMIC-1 climatology for residual ionospheric errors was introduced in the GEOS-IT DA experiments for Jan 2017. Temperature differences between the RIE-biased and the control DA results reveal a significant impact in the polar stratosphere with a bias as large as 2-4 K at 1 hPa and ~1 K at 10 hPa. There is noticeable (±0.3 K) temperature bias in the troposphere, which are likely caused by the indirect impact of assimilating the RO data. The diurnally varying RIEs do not appear to impact on the solar migrating tides in the analyzed temperatures in the upper stratosphere and

mesosphere, because the tidal waves are locked to the wave forcings from the lower atmosphere.

   The $\phi_{ex}$-gradient method requires the RO profile to reach at least 120 km so that the derived RIE is less sensitive to the sporadic-E (Es) layer influences. Es occurs mostly at 90-110 km with a strong localized effect, but it has a tailing effect in the RO $\phi_{ex}$ profile extending far below the Es layers before diminishing. Additional constraints from the RO measurements near 120 km and above would help substantially to provide a more accurate estimate of the RIE originated

from the F-region ionosphere.



RIEs were also found in the RO $\phi_{ex}$ measurements by comparing the profile with an expected exponential dependence. In addition, the RIE impacts can be seen in the small-scale variance of RO-derived temperature profile. In both cases, the RIE amplitudes appear to increase with height and is significant in the upper stratosphere. The findings from this study further emphasize the needs for accurate treatment of RIEs in future data assimilation and climate studies, especially with the growing infusion of commercial GNSS-RO data.

**Appendices**

**A. GNSS-RO Mission Summary**

The global GNSS-RO observations can be perhaps divided into three periods in terms of the total number of daily RO profiles: CHAMP-period (2001-2006), COSMIC1-period (2006-2019), COSMIC2-commercial-period (2019-). Advances in commercial RO receiver technologies play a critical role in the increase of the number of RO acquisitions from space in recent years. The BlackJack RO receiver on CHAMP, developed by NASA Jet Propulsion Laboratory (JPL), was able to track a dual-frequency GPS (G) signals for precise (cm accuracy range) orbit determination and continuous coverage [Hajj et al., 2004]. For the RO observation, the receiver software was also able to schedule high-rate (50 Hz) tracking of setting occultations of up to four GPS satellites. The BlackJack on CHAMP has only aft antennas for the RO sounding, which yielded ~250 profiles per day. The sampling was further improved by tracking rising occultations with the open-loop (OL) tracking successfully demonstrated on the SAC-C and later COMSIC-1 satellites. This advance led to the dual-antenna (aft and fore) and OL tracking IGOR (Integrated GPS and Occultation Receiver) implemented by the COMSIC-1 constellation [BRE, 2003], which produced ~700 daily ROs per satellite, or an average total of ~4000 daily ROs from ta six-satellite constellation.

The recent boost in the number of GNSS-RO observations came from availability of civil signals provided by GNSS satellites [GLONASS (R), Galileo (G), and BDS (C)], and from the combination of a new four-antenna TriG (Tri-band GNSS) receiver [Esterhuizen et al., 2009] on COSMIC-2 and commercial CubeSat constellations. While operational weather satellites such as MetOp [Zus et al., 2011] and FY3 [Bai et al., 2014] continue to provide global GNSS-RO observations, the commercial data from SmallSat/CubeSat constellations provided by Spire [Angling et al., 2021], GeoOptics [Chang et al., 2022], and PlanetiQ [Kursinski et al., 2021] have become increasingly important to yield the needed spatiotemporal coverage on the globe. The maximum RO top height, listed in Table A1 for these missions, is a key parameter to derive the RIE with the $\phi_{ex}$-gradient method presented in this study.

**Table A1.** Summary of GNSS-RO data used in this study

| LEO Satellites | Mission lifetime | Init,Final Alt (km) | Sun-syn (Asc ECT[1]) | Lat Coverage | Top RO Ht (km) | Tracked GNSS | Daily No. ROs |
|---|---|---|---|---|---|---|---|
| **CHAMP** | 2001-2008 | 450,330 | varying | 90°S/N | 140 | G | ~250 |





| COSMIC-1[2] | 2006-2020 | 525,810 | varying | 90°S/N | 130 | G | ~4000 |
|---|---|---|---|---|---|---|---|
| SAC-C | 2000-2013 | 705 | 10:30 | 90°S/N | 90 | G | 100-300 |
| MetOp-A[3] | 2006-2021 | 820 | 19:00 | 90°S/N | 90,300 | G | ~720 |
| MetOp-B[3] | 2012- | 820 | 19:00 | 90°S/N | 90,300 | G | ~700 |
| MetOp-C[3] | 2018- | 820 | 19:00 | 90°S/N | 90,300 | G | ~650 |
| C/NOPS | 2008-2015 | 850,350 | varying | 37°S/N | 170 | G | ~300 |
| KOMPSAT-5 | 2015- | 560 | 06:00 | 90°S/N | 135 | G | 300-600 |
| TSX | 2009- | 520 | 18:00 | 90°S/N | 135 | G | 150-300 |
| TDX | 2016- | 520 | 18:00 | 90°S/N | 135 | G | 150-300 |
| GRACE | 2007-2017 | 475,300 | varying | 90°S/N | 140 | G | 100-250 |
| FY-3C[4] | 2013- | 838,850 | 22:00 | 90°S/N | 130 | G | 400-550 |
| FY-3D | 2017- | 838 | 13:30 | 90°S/N | 130 | G | 400-600 |
| FY-3E[5] | 2021- | 830 | 05:30 | 90°S/N | 130 | G, C | ~1100 |
| FY-3G | 2023- | 414 | varying | 40°S/N | 130 | G, C, E | 1200-1600 |
| COSMIC-2[6] | 2019- | 715,540 | varying | 44°S/N | 90-500 | G,R | ~6500 |
| PAZ | 2018- | 520 | 18:00 | 90°S/N | 135 | G | 200-300 |
| Sentinel-6A | 2020- | 1336 | varying | 90°S/N | 80 | G,R | ~800 |
| GeoOptics | 2020-2022 | 490 | varying | 90°S/N | 145 | G | 300-1800 |
| Spire[7] | 2018- | 500-600 | varying | 90°S/N | 170-600 | G,R,E,J,C | ~4000[8] ~12,000[8] |
| PlanetiQ | 2023- | ~500 | varying | 90°S/N | 170 | G,R,E,C | 1000-3800 |

[1] Ascending-orbit equator crossing time (Asc ECT)

[2] The COSMIC1-3 spacecraft never reached the intended orbital altitude and was operated at 725km for the rest of its mission.

[3] Metop-A started to drift away from the Sun-sync orbit since ~2021. An extended RO experiment with Metop-A to acquire the high-rate data up to $h_t$=300km during 2020D161-2020D254. Following the successful experiment, the high-top RO acquisition has been implemented for the routine operation in MetOp-B/C since 2021.

[4] FY-3C started to drift away from the Sun-sync orbit (SSO) since 2016.

[5] FY-3E started to track GPS (G) and BDS (C), and FY-3G started to track GPS (G), BDS (C), and Galileo (E)

[6] The CDAAC COSMIC-2 NRT data contain GNSS-RO profiles from GPS (G) and GLONASS (R). The nominal RO top is ~140 km, but occasionally reaches up to 300 km or 500 km for space weather measurements.

[7] The Spire GNSS-RO observation tracks GPS (G), GLONASS (R), Galileo (E), and BDS (C) signals routinely to ~170 km, but for space weather observations the tracking often goes up to 300, 350, 500 or 600 km. The tracking of and QZSS (J) occurred briefly before 2021.

[8] NASA's Commercial Smallsat Data Acquisition (CSDA) data have a larger number of post-processing RO profiles per day from November 2019 to the present. The NOAA Commercial Data Program (CDP) acquires a fewer number of near-real-time (NRT) RO profiles per day, compared to the CSDA archive.







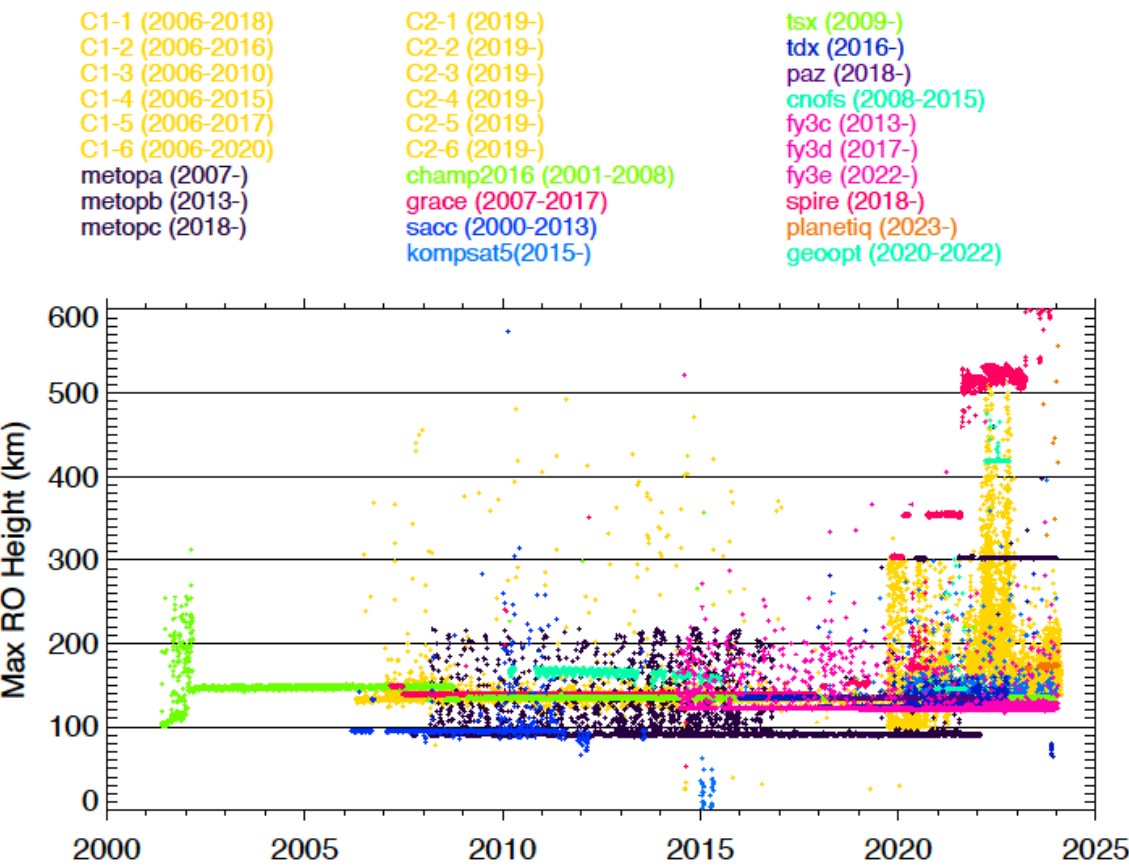

Fig.A1. Daily GNSS-RO statistics of the maximum RO top height from the past and current missions in 2001-2023.

**Competing interests**

The contact author has declared that none of the authors has any competing interests.

**Acknowledgements**

The work was by supported partially by NASA Goddard Space Flight Center (GSFC) Science Task Group (STG) fund and Commercial Smallsat Data Acquisition (CSDA) program. The authors thank UCAR COSMIC Data Analysis and Archive Center (CDAAC) and EUMETSAT Radio Occultation Meteorology Satellite Application Facility (ROM SAF) services for

data processing and distribution.



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
