# Peer review of "GNSS-RO Residual Ionospheric Error (RIE): A New Method and Assessment"

_Atmospheric Measurement Techniques, 2024_

## Referee Comment (RC1)

**GNSS-RO Residual Ionospheric Error (RIE): A New Method and Assessment**

Dong L. Wu et al., amt-2024-51, AMT-Review

General comment:

In general, I think the authors addressed a quite interesting and easy to implement approach for the correction of ionospheric residual errors in GNSS-RO data. They also provided a good literature overview, discussing the ongoing work and problems on this topic over the past years. Their style of writing was also good to follow, however, there are some technical errors/typos in this paper, which leave a bit of a sloppy impression. Furthermore, the paper is quite long and hence hard to read and concentrate on. I would prefer a clearer presentation of the main results, maybe providing some of the figures only as supplementary material. Personally, I appreciate the extensive analysis the authors conducted, however some of the information might get lost due to the length of the paper. They also add as an additional study the impact of these RIEs on data assimilation. By itself, this is of course interesting and important to discuss, however, I also feel they could have split the study maybe in two papers. To first introduce the method and precisely discuss the correction of RIEs on phase delays, and a second follow-up study with the data assimilation experiments. It reads more like a scientific report than a scientific publication, which should aim to concisely summarize and present the main/key findings. In that respect, I recommend the authors to improve the general style, structure, readability, and quality of the manuscript.

Furthermore, I wanted to address, that to my knowledge a correction on phase delays was already discussed in previous literature years ago, leading to the conclusion that a correction on bending angle is to be preferred. The problem here is that the dispersion residual (different ray paths, L1 and L2) is the most dominant residual, compared to higher-order ionospheric effects. Thereby, a correction on bending angles provides better results, since profiles are studied already on a common impact parameter, instead of on excess phase (see also Syndergaard 2000). Please provide a good and high-quality discussion on this issue. Readers should be aware of that, and understand why you don't see this as an issue and recommend this correction approach based on phase delays.

Summarized, I recommend a major revision in order to improve readability and a concise presentation of key results, and to get rid of most of the technical errors (I pointed out just a few, please re-check the complete paper carefully).

Specific comments:

L 61: ROPP is a processing package. So it is not "RO processing package **or** ROPP", better "a RO processing package **such as** ROPP"

L 71: "However, the GNSS-RO data infusion requires a key assumption about the $\alpha$ measurements in which ionospheric contributions can be fully removed by using the sounding from two L-band frequencies"; What is meant with "Infusion", what "key assumption". Please rephrase.

L 76: Please specify your statement "unrealistic". Why? I suggest dismissing this specific word.

L 102: I think there is a "minus-sign and absolute value" missing, $\alpha\_RIE = - |\kappa|(\alpha_1-\alpha_2)^2$... please check for the correct interpretation of this method.

L 106 to 109, 121-122: please provide a bracket around $(\alpha_1-\alpha_2)$ in the text.

L 112: Danzer et al. (2020) validated the kappa-corrected RO data against ERAint, ERA5, and MIPAS data. Please correct that statement. Furthermore, the warming was calculated solely based on RO, as a bias between RO-data with and without kappa-correction. The sentence reads wrong.

L 240: The RIE varies, as you state, with local time, season, solar cycle, solar activity, and RO receiver type. Maybe mention also geomagnetic term here. However, what I wanted to state, the bi-local correction is able to compute these variations. Please see, (i) Syndergaard and Kirchengast (2022) introducing the theory, and as application studies (ii) Liu et al. (2020): comparing kappa and bi-local as an initial study on bending angle, (iii) Liu et al. (2024): comparing kappa and bi-local on a larger scale also on temperature.

L 282: Related to that above statement, in Section 3.2, where a discussion is done based on a comparison with the kappa-correction, I suggest adding a discussion based on a comparison with the bi-local correction too. The bi-local correction can account for negative and positive biases resulting from including the geomagnetic term (see especially the analysis of Liu et al., 2024).

L 294 onwards: is this $(\Delta\alpha_1 - \Delta\alpha_2)^2$ a typo? I was confused. Shouldn't it be: $(\alpha_1 - \alpha_2)^2$? Please correct. If I am wrong, please make this clearer in the paper, and introduce the meaning properly. Thanks for this.

L326: You introduce $\sigma$ here for the first time, please make sure to introduce it already together with $\mu$ in line 320.

L 508: As you already address here, the second-order error can have positive and negative contributions. Please discuss it compared to the bi-local correction.

L545: This is an important conclusion: What about missions with a lower RO top height than 120km? Is this approach as a conclusion not recommended? Which missions does this concern? Please discuss. Further, what is the consequence for a complete re-processed multi-satellite data set (climatology), if this correction cannot be applied to all missions. Does this introduce a problem?

In general, in figures.

- Please provide units in a square bracket, e.g., [μrad], also for Latitude [°], solar local time [h], and so on…
- Also, the colorbars with 0.05x, or 0.07x are very confusing. Please provide a clearer solution here. Usually, one indicates the unit above or below the colorbar, and the range, which I guess means in your case 0.05 times the range from 0 to 5, might be adjusted, or the pre-factor added to the unit.
- What were the exact definitions for a "day" time and a "night" time window?
- Fig. 12: there is a strange offset in the colorbars.
- Fig. 20: increase x,y-labels.
- Please make sure that figure captions are located below the figure, and not land on the next page (see Fig. 1, Fig. 12, Fig. 20). At the beginning I thought they are completely missing. This helps the readability.

Technical corrections:

p. 2: Please remove the table of contents.

L 13: formulation is off, "therefore residual ionospheric error (RIE) is critical to accurately retrieve atmospheric temperature and refractivity"; reformulate

L 21: formulation "and in small-scale temperature variance of the RO retrieval". That is not a clear sentence.

L 27: Typo: "RIF"

L 101: introduced "the" so-called kappa-method

L110: delete the word "had", use instead "Liu et al. estimated"

L 141: "wehre"

L 142: "an RIE"

L 162: In the case „of" Fig. 1

L 209: Eq. 7: Bracket after the equation "…, with …"

L327: Please insert "commas" between a list of symbols such as $\Delta\alpha$, $\sigma$, $\mu$ and in general at several text places….

L 587: … range, like Es, as well as an extended...

L603: … greater **than** …

References, p. 36 onwards:

Please make sure that the references are given in a uniform way. For example, please compare the style in Angling et al. and Bai et al.

- Years are given after the list of names of the authors. Sometimes you put it at the end of the citation.
- Doi sometimes missing.
- Make sure that all links of the papers are imported as a link.

---

## Referee Comment (RC2)

**GNSS-RO Residual Ionospheric Error (RIE): A New Method and Assessment**

Dong L. Wu et al., amt-2024-51, AMT-Review

**General Comments**

1. This paper presents a new residual ionospheric errors (RIE) in bending angles based on the GNSS RO excess phase measurement for each RO event. The excess phase gradient method, is self-sufficient and based on the vertical derivative of the RO excess phase profile. Specifically, a linear fit was applied to the excess phase data at heights above 65 km, then calculate the RIE using the vertical derivative of the linear fit excess phase profile, finally the derived RIE is extrapolated to the RO measurements at the lower heights by assuming that $\Delta\alpha$ has the same impact on the entire $\alpha$ profile.

If I understand correctly, in this method the RIEs in bending angle are considered as the slopes of the linear fit excess phase profiles (as the red lines shown in the sub-figure (c) of figures 1-4). Then use this slopes as the RIE values for the entire bending angle profiles.

According to the sections "*2.1 Atmospheric Bending Angle ($\alpha$) and Excess Phase ($\phi ex$)*" and "*2.2 RIE and Detection Method*" this mothed has 3 assumptions:

(1) "*For a rising/setting occultation, $V\perp$ is the ascending/descending rate of RO sampling with respect to ht, or the GNSS–LEO straight line height (SLH), which yields $V\perp \cong dht / dt$. The get equation (6).*" **Which uses the $V\perp$ of the LEO satellite as the tangent point velocity. In the GNSS-LEO RO, this assumption will induce errors.**

(2) "*In the upper atmosphere where there is little atmospheric bending (i.e., $\alpha c \approx 0$), a significant value that is not zero in $d\phi/dht$ ( indicates the existence of $\alpha RIE$, which can be both positive and negative.*" **Define the $\alpha$ calculated by equations (4) and (6) as bending angle RIEs. Actually, the equations (4) and (6) calculate the ionospheric bending angles above ~80 km, physically this variable is different from the bending angle RIEs defined in the previous studies.**

(3) The equation (6) is used for the linear fit excess phase profiles (as the red lines shown in the sub-figure (c) of figures 1-4). Then use this slopes as the RIE values for the entire bending angle profiles. **As discussed in the manuscript, the fit excess phase profiles depend on the local time, season, solar cycle, solar activity, and RO receiver type, RO top height. Maybe also geomagnetic field, the RO plane direction and so on. While this method only use equation (6) to calculate the ionospheric bending angles as bending angle RIE. This will induce problems in the application.**

2. Regarding the quality control (QC) on the excess phase data as shown in Table 1, how to determine the QC flags and thresholds? It does not according to the previous bending angle RIE definition and characteristics. To "Retain only realistic $\Delta\alpha$ values", set $|\Delta\alpha| < 2000$ μrad,

this threshold is too large. (As shown in your figures, most of the $|\Delta\alpha|$ are less than 2 μrad).

3. Regarding the $\Delta\alpha$ statistics with the latitude: Figures 5-9 show that most of the $\Delta\alpha$ values for day and night from Jan 2013 are positive, while Figure 19 shows most of the $\Delta\alpha$ values for day and night MetOp RO data from 2020 are negative. Why?
It also shows that this mothed is very sensitive with the RO top height. When the height increases the ionospheric bending angle will become larger and non-linear, this may be a reason.

4. Regarding the $\Delta\alpha$ statistics with the local time: As shown in figure 10, the $\Delta\alpha$ statistical behaviors are very strange (not reasonable). (1) from -60 to 60 latitude degree, at local time 8 and 20, where the ionosphere has large horizontal gradient since the morning and dusk change, and the magnitude of the RIEs are very large, however in figure 10 in this area the $\Delta\alpha$ is around zero. (2) the $\Delta\alpha$ magnitudes at night time are larger than the daytime. (3) generally, the night time RIEs are near zero, however in figure 10 they relatively larger than that in the daytime and with positive sign, which indicated that the positive $\Delta\alpha$ values in Figures 5-9 mainly come from the night time data.

5. As this is a new method and can be used for each individual RO profile, therefore it's better to show the profile-by-profile RIEs and their vertical statistical variables of biases and stdev, which is easier for readers to understand the results, also easier for comparing with previous studies.

Specific comments:
Please update the figures by providing proper units, using uniform color bars in one figure. It's better to combine the same layout figures like figures 1-4 into one figure, since there are so many figures in this paper.

There are lots of typos in the manuscript, please revise them, for examples:
L27: "RIF"
L141: "wehre"
L406: "(2),"
…

---

## Author Comment (AC1)

Response to Review#2

**General Comments**

1. This paper presents a new residual ionospheric errors (RIE) in bending angles based on the GNSS RO excess phase measurement for each RO event. The excess phase gradient method, is self-sufficient and based on the vertical derivative of the RO excess phase profile. Specifically, a linear fit was applied to the excess phase data at heights above 65 km, then calculate the RIE using the vertical derivative of the linear fit excess phase profile, finally the derived RIE is extrapolated to the RO measurements at the lower heights by assuming that $\Delta\alpha$ has the same impact on the entire $\alpha$ profile.

If I understand correctly, in this method the RIEs in bending angle are considered as the slopes of the linear fit excess phase profiles (as the red lines shown in the sub-figure (c) of figures 1-4). Then use this slopes as the RIE values for the entire bending angle profiles.

Your understanding is correct.

According to the sections "*2.1 Atmospheric Bending Angle ($\alpha$) and Excess Phase ($\phi ex$)*" and "*2.2 RIE and Detection Method*" this mothed has 3 assumptions:

- *"For a rising/setting occultation, $V\perp$ is the ascending/descending rate of RO sampling with respect to ht, or the GNSS–LEO straight line height (SLH), which yields $V\perp \cong dht/dt$. The get equation (6)."* **Which uses the $V\perp$ of the LEO satellite as the tangent point velocity. In the GNSS-LEO RO, this assumption will induce errors.**

This assumption works well at ht > 30 km where the $V\perp$ is close to a constant and the error is small compared to RIE. In the lower atmosphere where the bending is severe, $V\perp$ is no longer constant with respect to ht and an inversion is required to determine the bending.

From the review comments, we feel that one of the key points in this paper was not well communicated. Therefore, we added Appendix A to discuss how RIEs can arise in

the case without bending. It's a misconception to attribute RIEs solely to the bending effect.

Appendix A provide more discussions on 'bending delay and phase advance' from radio wave propagation in plasma. Especially, the phase advance due to the faster-than-light phase velocity from propagation in plasma can be mistakenly interpreted as a bending. In fact, it is an independent effect from bending (due to group velocity) in the GNSS-RO excess phase measurement. This is also the major reason that this study argues to analyze the excess phase data, rather the bending data, of which the latter would mislead what might cause the RIE. In Appendix A, we discuss the situation that RIEs can occur even without bending.

- *"In the upper atmosphere where there is little atmospheric bending (i.e., $\alpha c \approx 0$), a significant value that is not zero in $d\phi/dht$ ( indicates the existence of $\alpha RIE$, which can be both positive and negative."* **Define the $\alpha$ calculated by equations (4) and (6) as bending angle RIEs. Actually, the equations (4) and (6) calculate the ionospheric bending angles above ~80 km, physically this variable is different from the bending angle RIEs defined in the previous studies.**

See the above for explanation. We believe that RIEs contain errors other than those from the bending effect. Therefore, it is more appropriate to characterize RIEs using the excess phase measurements, rather the bending angle.

Because the bending angle has been used to compare the amplitudes of RIE derived from different methods, here in this study we adopt the equation of bending angle expression but do not fully agree with the bending as the sole contribution to RIEs.

- The equation (6) is used for the linear fit excess phase profiles (as the red lines shown in the sub-figure (c) of figures 1-4). Then use this slopes as the RIE values for the entire bending angle profiles. **As discussed in the manuscript, the fit excess phase profiles depend on the local time, season, solar cycle, solar activity, and RO receiver type, RO top height. Maybe also geomagnetic field, the RO plane direction and so on. While this method only use equation (6) to calculate the ionospheric bending angles as bending angle RIE. This will induce problems in the application.**

The fitting does not require any knowledge about local time, season, solar cycle, solar activity, etc. The results from the fitting do, which is called the RIE in this study.

Also in the revised manuscript we made it clear that interpreting the vertical gradient of excess phase profile as a bending would be misleading since a RIE can occur even without bending.

1. Regarding the quality control (QC) on the excess phase data as shown in Table 1, how to determine the QC flags and thresholds? It does not according to the previous bending angle RIE definition and characteristics. To "Retain only realistic $\Delta\alpha$ values", set $|\Delta\alpha|$ < 2000 µrad, this threshold is too large. (As shown in your figures, most of the $|\Delta\alpha|$ are less than 2 µrad).

It was a typo. It should be 2 µrad and has been corrected in the revision.

1. Regarding the $\Delta\alpha$ statistics with the latitude: Figures 5-9 show that most of the $\Delta\alpha$ values for day and night from Jan 2013 are positive, while Figure 19 shows most of the $\Delta\alpha$ values for day and night MetOp RO data from 2020 are negative. Why?

Thank you for catching this. It was a plotting error in Fig.19 and has been corrected in the revision.

It also shows that this mothed is very sensitive with the RO top height. When the height increases the ionospheric bending angle will become larger and non-linear, this may be a reason.

It is sensitive to the RO top height up to a certain altitude, which is largely due to sporadic-E (Es) related perturbations. Es often induces a large oscillation at 80-100 km, which can influence the fitting substantially. There is essentially little way to get around these perturbations if the RO profile is cut off around 85 km. Above 110km, fortunately, Es-induced oscillations are small, allowing the fitting method to establish a more robust estimate of the RIE.

In the revised manuscript we also pointed out that Es tends to have a tailing effect below 80km. But it reduces sharply with height as the RO sounding goes below the Es layer. As revealed in other studies, the Es effect is evident in the iono-free bending angle profile as well.

The new method proposed in this study aims to capture the RIEs induced by the F-region ionosphere and above, not by the Es layers, since those errors may have an extended impact on the RO sounding of the atmosphere.

1. Regarding the $\Delta\alpha$ statistics with the local time: As shown in figure 10, the $\Delta\alpha$ statistical behaviors are very strange (not reasonable). (1) from -60 to 60 latitude degree, at local time 8 and 20, where the ionosphere has large horizontal

gradient since the morning and dusk change, and the magnitude of the RIEs are very large, however in figure 10 in this area the $\Delta\alpha$ is around zero. (2) the $\Delta\alpha$ magnitudes at night time are larger than the daytime. (3) generally, the night time RIEs are near zero, however in figure 10 they relatively larger than that in the daytime and with positive sign, which indicated that the positive $\Delta\alpha$ values in Figures 5-9 mainly come from the night time data.

Again, we would not consider that every RIE be induced by the bending effect. We believe that RIEs can come from the radio wave propagation without bending. Appendix A illustrates a simplistic scenario for no bending propagation. In reality, the L1 and L2 bands may split their propagating paths at a location with small-scale inhomogeneous structures and continue with their journey different through the ionosphere. This type of ionospheric propagation could have a small or little bending, but producing a large amplitude of RIE in excess phase from the phase advance differences.

1. As this is a new method and can be used for each individual RO profile, therefore it's better to show the profile-by-profile RIEs and their vertical statistical variables of biases and stdev, which is easier for readers to understand the results, also easier for comparing with previous studies.

In Figs.1-4 (panel c), we provided the fitted slope for each profile example. These examples also highlight the challenges to infer the slope in the presence of large oscillations in the excess phase measurements. Therefore, the inferred slopes are expected to have a large standard deviation that is mainly due to the oscillatory nature of excess phase data. The PDF plots in Figs.5-9 were intended to show the spread of RIEs from the fitting. Fig.10 provided the values of RIE standard deviation as a function of latitude and local time.

Specific comments:

Please update the figures by providing proper units, using uniform color bars in one figure. It's better to combine the same layout figures like figures 1-4 into one figure, since there are so many figures in this paper.

There are lots of typos in the manuscript, please revise them, for examples:

L27: "RIF"

L141: "wehre"

L406: "(2),"

We have corrected these typos among others in the revised manuscript.

---

## Author Response (AR1)

Response to Review#1

General comments

In general, I think the authors addressed a quite interesting and easy to implement approach for the correction of ionospheric residual errors in GNSS-RO data. They also provided a good literature overview, discussing the ongoing work and problems on this topic over the past years. Their style of writing was also good to follow, however, there are some technical errors/typos in this paper, which leave a bit of a sloppy impression. Furthermore, the paper is quite long and hence hard to read and concentrate on. I would prefer a clearer presentation of the main results, maybe providing some of the figures only as supplementary material.

We revised the paper considerably to take care typos and English. We added Appendix A to provide more discussions on 'bending delay and phase advance' for radio wave propagation in plasma.

Personally, I appreciate the extensive analysis the authors conducted, however some of the information might get lost due to the length of the paper. They also add as an additional study the impact of these RIEs on data assimilation. By itself, this is of course interesting and important to discuss, however, I also feel they could have split the study maybe in two papers.

To first introduce the method and precisely discuss the correction of RIEs on phase delays, and a second follow-up study with the data assimilation experiments. It reads more like a scientific report than a scientific publication, which should aim to concisely summarize and present the main/key findings. In that respect, I recommend the authors to improve the general style, structure, readability, and quality of the manuscript.

We moved a large part of the DA impact discussions to Appendix B, and keep the key results and summary in the main section.

Furthermore, I wanted to address, that to my knowledge a correction on phase delays was already discussed in previous literature years ago, leading to the conclusion that a correction on bending angle is to be preferred. The problem here is that the dispersion residual (different ray paths, L1 and L2) is the most dominant residual, compared to higher-order ionospheric effects. Thereby, a correction on bending angles provides better results, since profiles are studied already on a common impact parameter, instead of on excess phase (see also Syndergaard 2000). Please provide a good and high-quality discussion on this issue. Readers should be aware of that, and understand why you don't see this as an issue and recommend this correction approach based on phase delays.

From the review comments, we feel that one of the key points in this paper was not well communicated. Therefore, we added Appendix A to discuss how RIEs can arise in the case without bending. It's a misconception to attribute RIEs solely to the bending effect.

Appendix A provide more discussions on 'bending delay and phase advance' from radio wave propagation in plasma. Especially, the phase advance due to the faster-than-light phase velocity from propagation in plasma can be mistakenly interpreted as a bending. In fact, it is an independent effect from bending (due to group velocity) in the GNSS-RO excess phase measurement. This is also the major reason that this study argues to analyze the excess phase data, rather the bending data, of which the

latter would mislead what might cause the RIE. In Appendix A, we discuss the situation that RIEs can occur even without bending.

Summarized, I recommend a major revision in order to improve readability and a concise presentation of key results, and to get rid of most of the technical errors (I pointed out just a few, please re-check the complete paper carefully).

The manuscript has been revised to take these advices in consideration.

Specific comments:

L 61: ROPP is a processing package. So it is not "RO processing package **or** ROPP", better "a RO processing package **such as** ROPP"

Correction was made in the revision as suggested.

L 71: "However, the GNSS-RO data infusion requires a key assumption about the $\alpha$ _measurements in which ionospheric contributions can be fully removed by using the sounding from two L-band frequencies"; What is meant with "Infusion", what "key assumption". Please rephrase.

The sentence was modified as:

"However, the benefit of GNSS-RO data in DA requires ionospheric contributions to be fully removed for the $\alpha$ measurements."

L 76: Please specify your statement "unrealistic". Why? I suggest dismissing this specific word.

We would like to emphasize the day-night difference in the solar-cycle variations in the bending angle. The sentence was modified as:

"For example, Danzer et al. [2013] highlighted an unrealistic solar cycle variation induced by the daytime ionosphere in the simulated atmospheric bending angle."

L 102: I think there is a "minus-sign and absolute value" missing, $\alpha\_RIE = - |\kappa|(\alpha1-\alpha2)2...$ please check for the correct interpretation of this method.

This has been corrected, along with the sentence that describes this expression.

L 106 to 109, 121-122: please provide a bracket around $(\alpha1-\alpha2)$ in the text.

Changed accordingly.

L 112: Danzer et al. (2020) validated the kappa-corrected RO data against ERAint, ERA5, and MIPAS data. Please correct that statement. Furthermore, the warming was calculated solely based on RO, as a bias between RO-data with and without kappa-correction. The sentence reads wrong.

Changed accordingly. The new sentences read as follows:

"The $\kappa$ model predicts a lower RIE value during the daytime and higher F10.7. Danzer et al. [2020] further validated the $\kappa$-model for RIE correction with the European Center for Medium-range Weather Forecast reanalysis (ERA-Interim, Dee et al., 2011; and ERA5, Hersbach et al., 2020), reporting warming (0.2 – 2 K) effects at 40-45 km prior to the $\kappa$-model correction (0.01-0.05 μrad). Using a different model, so-called bi-local correction approach, Liu et al. [2020] showed that the $\alpha_{RIE}$ values are comparable to the $\kappa$-model with an amplitude < 0.05 μrad but the standard deviation of $\alpha_{RIE}$ is larger than its mean at all heights."

L 240: The RIE varies, as you state, with local time, season, solar cycle, solar activity, and RO receiver type. Maybe mention also geomagnetic term here. However, what I wanted to state, the bi-local correction is able to compute these variations. Please see, (i) Syndergaard and Kirchengast (2022) introducing the theory, and as application studies (ii) Liu et al. (2020): comparing kappa and bi-local as an initial study on bending angle, (iii) Liu et al. (2024): comparing kappa and bi-local on a larger scale also on temperature.

We have included a brief review on the magnetic field impact in the introduction, as well as the papers by Syndergaard and Kirchengast (2022) for the 3D effect and Liu et al. (2020) for bi-local modeling. We can't find the reference Liu et al. (2024) to comment on the k-method and bi-local comparisons. We did observe and cited the similar RIEs amplitudes [Liu et al., 2020, Fig.5 therein] between the two approaches, which showed mostly negative RIE values.

L 282: Related to that above statement, in Section 3.2, where a discussion is done based on a comparison with the kappa-correction, I suggest adding a discussion based on a comparison with the bi-local correction too. The bi-local correction can account for negative and positive biases resulting from including the geomagnetic term (see especially the analysis of Liu et al., 2024).

We provided a discussion on potential geomagnetic effects in section 3.3 with Fig.13. We found a weak dependence on B-field but a stronger connection to sporadic-E (Es). The latter is perhaps related to the 3D effect in the calculation outlined by Syndergaard and

Kirchengast (2022) who divided the ray trace model into the near and far side of the tangent point. If Es splits the L1 and L2 paths at the tangent point, the RIE would arise due to the near-side propagation both from phase advance in plasma and phase delay in bending.

We added more discussions in section 3.3 on this point.

[Figure]

Fig.13. Geographical maps of the $\Delta\alpha$ derived from COSMIC-1 $-d\phi_{exL1}/dh_t$ measurements for January and July 2013; the white lines display positions of the geomagnetic equator.

L 294 onwards: is this (Δα1- Δα2)2 a typo? I was confused. Shouldn't it be: (α1- α2)2? Please correct. If I am wrong, please make this clearer in the paper, and introduce the meaning properly. Thanks for this.

This is correct.

In this paper we try to reserve $\alpha$ for conventional definition of bending angle and $\Delta\alpha$ as an approximation of $\alpha$ from the vertical derivative of excess phase profile.

We made this clear right after Eq.7 where $\Delta\alpha$ is introduced.

L326: You introduce σ here for the first time, please make sure to introduce it already together with μ in line 320.

Done.

L 508: As you already address here, the second-order error can have positive and negative contributions. Please discuss it compared to the bi-local correction.

We added more discussions in section 3.3 on this point.

L545: This is an important conclusion: What about missions with a lower RO top height than 120km? Is this approach as a conclusion not recommended? Which missions does this concern? Please discuss. Further, what is the consequence for a complete re-processed multi-satellite data set (climatology), if this correction cannot be applied to all missions. Does this introduce a problem?

We are troubled by these missions as well.

Shortly after we identified the importance of Es in GNSS-RO [Wu et al., 2005], we recommended to all operators to raise the RO top height to >120 km. But Metop, Kompsat, TSX and TDX, and FY3C/D among a few, did not change their operation. Recently, Metop and FY3 have raised the RO profile top in the normal operation.

For those missions with a lower RO top, it would require a climatology built upon other missions that can be parameterized as a function of latitude, longitude, local time and solar cycle.

We added this in the discussion section.

In general, in figures.

- Please provide units in a square bracket, e.g., [μrad], also for Latitude [°], solar local time [h], and so on...

This change would require a lot of rework on the figures made previously. Instead, we made it clear that all variable units are consistent in all figures.

- Also, the colorbars with 0.05x, or 0.07x are very confusing. Please provide a clearer solution here. Usually, one indicates the unit above or below the colorbar, and the range, which I guess means in your case 0.05 times the range from 0 to 5, might be adjusted, or the pre-factor added to the unit.

We added the following clarification in the figure capture:

"All color numbers have a scale factor indicated at the top of each colorbar, and has the variable unit indicated in the () bracket."

- What were the exact definitions for a "day" time and a "night" time window?

We used the solar zenith angle 90 degrees to separate day and night. A clarification is made in the revision.

- Fig. 12: there is a strange offset in the colorbars.

As shown in the time series, the January RIE is generally larger than the July, and both are larger than those from the equinoxes. The colorbars are scaled differently to account for these differences.

- Fig. 20: increase x,y-labels.

The font size is increased in the revision.

- Please make sure that figure captions are located below the figure, and not land on the next page (see Fig. 1, Fig. 12, Fig. 20). At the beginning I thought they are completely missing. This helps the readability.

We will make sure that this shows well in the final print.

Technical corrections:

p. 2: Please remove the table of contents.

Done!

L 13: formulation is off, "therefore residual ionospheric error (RIE) is critical to accurately retrieve atmospheric temperature and refractivity"; reformulate

It is re-phrased as follows:

" Because the magnitudes of the RO bending angle are small at these altitudes, quantifying and removing residual ionospheric error (RIE) are critical to accurately retrieve atmospheric temperature and refractivity."

L 21: formulation "and in small-scale temperature variance of the RO retrieval". That is not a clear sentence.

In the revised manuscript, it is put in a separate sentence:

"RIEs are likely to impact the RO temperature retrieval by inducing a small-scale variance that is solar-cycle dependent."

L 27: Typo: "RIF"

Corrected.

L 101: introduced "the" so-called kappa-method

Corrected.

L110: delete the word "had", use instead "Liu et al. estimated"

Corrected.

L 141: "wehre"

Corrected.

L 142: "an RIE"

Corrected.

L 162: In the case „of" Fig. 1

Corrected.

L 209: Eq. 7: Bracket after the equation "..., with ..."

Corrected.

L327: Please insert "commas" between a list of symbols such as Δα, σ, μ and in general at several text places....

Corrected.

L 587: ... range, like Es, as well as an extended...

Corrected.

L603: ... greater **than** ...

Corrected.

References, p. 36 onwards:

Please make sure that the references are given in a uniform way. For example, please compare the style in Angling et al. and Bai et al.

- Years are given after the list of names of the authors. Sometimes you put it at the end of the citation.

- Doi sometimes missing.

- Make sure that all links of the papers are imported as a link.

Corrections are made.

---

## Referee Report (RR1)

**GNSS-RO Residual Ionospheric Error (RIE):**
**A New Method and Assessment**

Dong L. Wu et al., amt-2024-51, AMT-Review, Round 2: 26-09-2024

General comments:

In general, the authors addressed finally all questions. (At the beginning not all answers were provided at the AMT platform. Maybe an upload mistake?) However, when I checked for the **proposed modifications**, **I partially couldn't find them in the manuscript**. I am not sure what has happened. **Maybe the wrong track-changes file was uploaded?** Maybe I made a mistake and downloaded the wrong file. But I suggest to the authors to carefully go through the manuscript and check, if everything is included.

**Further remark:** After line 240, I stopped comparing if everything is included. I expect the authors to do that.

**Please revise the manuscript carefully** and provide the correct track-changes version.

A few specific examples and comments:

**My old comment:** L 71: "However, the GNSS-RO data infusion requires a key assumption about the $\alpha$ measurements in which ionospheric contributions can be fully removed by using the sounding from two L-band frequencies"; What is meant with "Infusion", what "key assumption". Please rephrase.

**Answers from the authors:** The sentence was modified as:

"However, the benefit of GNSS-RO data in DA requires ionospheric contributions to be fully removed for the $\alpha$ measurements."

**The correction was not made in the manuscript. Still the old formulation included.**

**Former comment on L 76:** Please specify your statement "unrealistic". Why? I suggest dismissing this word.

**Answers from the authors:** We would like to emphasize the day-night difference in the solar-cycle variations bending angle. The sentence was modified as:

"For example, Danzer et al. [2013] highlighted an unrealistic solar cycle variation by the daytime ionosphere in the simulated atmospheric bending angle."

I want to address once more my question here: "**why unrealistic**". It was a study directly performed on RO profiles, calculating the bending angle bias, and a further study with simulated data using NeUoG. The observed RO bending angle bias and simulated RO bending angle bias overlap. Furthermore, the F10.7 index was rather high in 2001/2002 years. Please remove "unrealistic" and soften the wording, such as "For example, Danzer et al. (2013) observe a rather high solar cycle

variation by the daytime ionosphere in the simulated and observed atmospheric bending angle bias."

**My old comment:** L 102: I think there is a "minus-sign and absolute value" missing, $\alpha\_RIE = - |\kappa|(\alpha1-\alpha2)2$... please check for the correct interpretation of this method.

**Answers from the authors:** This has been corrected, along with the sentence that describes this expression.

**There is still no minus sign in the equation.**

**My old comment:** L 106 to 109, 121-122: please provide a bracket around $(\alpha1-\alpha2)$ in the text.

**Answers from the authors:** This Changed accordingly.

**There is still not always a bracket around ($\alpha1-\alpha2$)**

Further comments

L 149: as 'a' misconception

L153: "higher-order"

**My old comment:** L 240: The RIE varies, as you state, with local time, season, solar cycle, solar activity, and RO receiver type. Maybe mention also geomagnetic term here. However, what I wanted to state, the bi-local correction is able to compute these variations. Please see, (i) Syndergaard and Kirchengast (2022) introducing the theory, and as application studies (ii) Liu et al. (2020): comparing kappa and bi-local as an initial study on bending angle, (iii) Liu et al. (2024): comparing kappa and bi-local on a larger scale also on temperature.

**Answers from the authors:** We have included a brief review on the magnetic field impact in the introduction, as well as the papers by Syndergaard and Kirchengast (2022) for the 3D eUect and Liu et al. (2020) for bi-local modeling. We can't find the reference Liu et al. (2024) to comment on the k-method and bi-local comparisons. We did observe and cited the similar RIEs amplitudes [Liu et al., 2020, Fig.5 therein] between the two approaches, which showed mostly negative RIE values.

**Where is the discussion. I couldn't find it.**

Here is the reference:

Liu, C., Danzer, J., Kirchengast, G., Haas, S. J., Proschek, V., Schwaerz, M., ... & Wang, X. (2024). Understanding ionospheric and geomagnetic effects on residual biases in radio occultation data for stratospheric climate monitoring. Journal of Geophysical Research: Space Physics, 129(5), e2023JA032110.

In general, in figures.

**My old comment:** Please provide units in a square bracket, e.g., [μrad], also for Latitude [°], solar local time [h], and so on…

**Answers from the authors:** This change would require a lot of rework on the figures made previously. Instead, we made it clear that all variable units are consistent in all figures.

But this is not correct. **Units are supposed to be in a square bracket**. Otherwise, you would read it as an equation. Please correct!

**My old comment:** References:

Please make sure that the references are given in a uniform way. For example, please compare the style in Angling et al. and Bai et al.

- Years are given after the list of names of the authors. Sometimes you put it at the end of the citation.

- Doi sometimes missing.

- Make sure that all links of the papers are imported as a link.

**Answers from the authors:** Corrections are made.

**There are still not all references consistent.** E.g., 871 and others

L877: Wu is in bold.

---

## Author Response (AR2)

Response to the 2$^{nd}$ Report from Review#1

General comments

General comments: In general, the authors addressed finally all questions. (At the beginning not all answers were provided at the AMT platform. Maybe an upload mistake?) However, when I checked for the proposed modifications, I partially couldn't find them in the manuscript. I am not sure what has happened. Maybe the wrong track-changes file was uploaded? Maybe I made a mistake and downloaded the wrong file. But I suggest to the authors to carefully go through the manuscript and check, if everything is included. Further remark: After line 240, I stopped comparing if everything is included. I expect the authors to do that. Please revise the manuscript carefully and provide the correct track-changes version.

From your comments, it doesn't seem that you have seen the correct version of revised manuscript, because the revisions described in our response were incorporated.

My old comment: L 71: "However, the GNSS-RO data infusion requires a key assumption about the $\alpha$ measurements in which ionospheric contributions can be fully removed by using the sounding from two L-band frequencies"; What is meant with "Infusion", what "key assumption". Please rephrase.

Answers from the authors: The sentence was modified as: "However, the benefit of GNSS-RO data in DA requires ionospheric contributions to be fully removed for the $\alpha$ measurements."

 **The correction was not made in the manuscript. Still the old formulation included.**

By 'key assumption' we meant that the $\alpha$ measurements contain no RIE and all ionospheric contributions can be fully removed with a linear combination of the measurements from two L-band frequencies'. Here is the new sentence in the revision:

"However, the benefit of GNSS-RO data in DA requires that the $\alpha$ measurements contain no RIE and all ionospheric contributions can be fully removed with a linear combination of the measurements from two L-band frequencies."

Former comment on L 76: Please specify your statement "unrealistic". Why? I suggest dismissing this word. Answers from the authors: We would like to emphasize the day-night difference in the solar-cycle variations bending angle.

The sentence was modified as:
"For example, Danzer et al. [2013] highlighted an unrealistic solar cycle variation by the daytime ionosphere in the simulated atmospheric bending angle."

I want to address once more my question here: "why unrealistic". It was a study directly performed on RO profiles, calculating the bending angle bias, and a further study with simulated data using NeUoG. The observed RO bending angle bias and simulated RO bending angle bias overlap. Furthermore, the F10.7 index was rather high in 2001/2002 years. Please remove "unrealistic" and soften the wording, such as "For example, Danzer et al. (2013) observe a rather high solar cycle

variation by the daytime ionosphere in the simulated and observed atmospheric bending angle bias."

We softened the statement by removing 'unrealistic' in the new revision.

My old comment: L 102: I think there is a "minus-sign and absolute value" missing, $\alpha\_RIE = - |\kappa|(\alpha1-\alpha2)2$… please check for the correct interpretation of this method.

Answers from the authors: This has been corrected, along with the sentence that describes this expression.

There is still no minus sign in the equation.

We verified this. The track-change file might be from an old revision. But it was corrected in the clean version submitted. Anyhow, the corrected version is uploaded this time.

My old comment: L 106 to 109, 121-122: please provide a bracket around $(\alpha1-\alpha2)$ in the text.

Answers from the authors: This Changed accordingly.

There is still not always a bracket around $(\alpha1-\alpha2)$

We found a few places and made the correction as suggested.

L 149: as 'a' misconception

Corrected.

L153: "higher-order" My old comment: L 240: The RIE varies, as you state, with local time, season, solar cycle, solar activity, and RO receiver type. Maybe mention also geomagnetic term here. However, what I wanted to state, the bi-local correction is able to compute these variations. Please see, (i) Syndergaard and Kirchengast (2022) introducing the theory, and as application studies (ii) Liu et al. (2020): comparing kappa and bi-local as an initial study on bending angle, (iii) Liu et al. (2024): comparing kappa and bi-local on a larger scale also on temperature.

Answers from the authors: We have included a brief review on the magnetic field impact in the introduction, as well as the papers by Syndergaard and Kirchengast (2022) for the 3D eUect and Liu et al. (2020) for bi-local modeling. We can't find the reference Liu et al. (2024) to comment on the k-method and bi-local comparisons. We did observe and cited the similar RIEs amplitudes [Liu et al., 2020, Fig.5 therein] between the two approaches, which showed mostly negative RIE values.

Where is the discussion. I couldn't find it.

We had the following discussion (red text) in section 3.3 from the previous revision:

However, it remains unclear to what extent Es may contribute to the RIE amplitude and variability. Although the $d\phi_{ex}/dh_t$ method attempts to minimize the Es impacts using more measurements from higher altitudes [Fig.2], the RIE maps from Fig.13 seem to indicate that Es may have a significant role in the nighttime RIE variation. The fact that $d\phi_{ex}/dh_t$ is correlated more to Es than to the geomagnetic field suggests that the spatial inhomogeneity effect might play a significant role in RIE. As described by Syndergaard and Kirchengast (2022) in a bi-local ray trace model, an RIE would arise due the L1 and L2 path split at the tangent point [Appendix A]. Most of the contribution to RIE comes from the near-side propagation after the split, where the L1 and L2 phase advance (in plasma propagation) and phase delay (from F-region bending) can go through significantly different paths. Because the E- and F-region ionospheric variabilities are driven by different processes, their contributions to the RIE may depend on latitude, longitude, local time and geomagnetic field. As elucidated by Syndergaard and Kirchengast [2022], path differences between the L1 and L2 propagation in a 3D structured ionosphere are the major cause of various RIEs, which can vary with the geomagnetic field and the spatial distribution and gradient of electron density. However, in a comparison between the simulated bi-local and $\kappa$-model RIEs, Liu et al. [2024] found a significant geomagnetic impact through high-order contributions to the refractive index but no significant effect from ionospheric asymmetry. One possibility of the negligible impact from ionospheric asymmetry in the ray-trace simulations by Liu et al. [2024] is the way how the asymmetry was incorporated in the model. In the study by Liu et al. [2024], an asymmetry factor was induced to partition the vertical TEC (vTEC) on the near and far-side ionosphere divided at the tangent point. This is likely a different inhomogeneity from the propagation path split implied by Syndergaard and Kirchengast (2022). It would require a strong vertical gradient in *Ne* such as Es to split the propagation paths between L1 and L2. The vTEC partitioning approach implemented by Liu et al. [2024] may not induce extraordinarily strong vertical *Ne* gradient in the inhomogeneous ionosphere to test the impacts from the case with fine structures. Hence, depending on the relative importance of these contributions, the RIE correction methods are likely to yield different impacts on the neutral atmospheric measurements.

Here is the reference:

Liu, C., Danzer, J., Kirchengast, G., Haas, S. J., Proschek, V., Schwaerz, M., ... & Wang, X. (2024). Understanding ionospheric and geomagnetic effects on residual biases in radio occultation data for stratospheric climate monitoring. Journal of Geophysical Research: Space Physics, 129(5), e2023JA032110.

Thank you for the reference. We included more discussions on the relative importance of geomagnetic effects and propagation path differences in contributing to RIE. The new discussion on the study by Liu et al. [2024] is in green.

In general, in figures. My old comment: Please provide units in a square bracket, e.g., [μrad], also for Latitude [°], solar local time [h], and so on...

Answers from the authors: This change would require a lot of rework on the figures made previously. Instead, we made it clear that all variable units are consistent in all figures.

But this is not correct. Units are supposed to be in a square bracket. Otherwise, you would read it as an equation. Please correct!

We made a great effort to change the brackets in all figures as suggested.

My old comment: References: Please make sure that the references are given in a uniform way. For example, please compare the style in Angling et al. and Bai et al. - Years are given after the list of names of the authors. Sometimes you put it at the end of the citation. - Doi sometimes missing. - Make sure that all links of the papers are imported as a link.

Answers from the authors: Corrections are made.

There are still not all references consistent. E.g., 871 and others

L877: Wu is in bold.

All references are checked for consistency. The missing doi has been added if it is available.

---

## Author Response (AR3)

Response to Editor's comments (in color):

It seems that the green part of the response file (section 3.3) is neither included in the manuscript nor in the track-changes file.
The reference to Liu et al. (2024) is included in the reference list, but the discussion about this paper is still missing in these files.

Thank you for spotting this error. For some reason, the cut-and-paste from our responses didn't come through. This error has been corrected in the new version uploaded.

Technical error:
Line 57: dual-frequency first order ($f$ ) correction

The empty square is now removed.

The quality of the figures could be improved, e.g. some have now units in a square bracket, but they seem just to be pasted
on top of the original figures - using different fonts. However, I will leave it to the copy-editor to decide, if this needs to be changed.

We did make a change in the unit format with square backets for all figures by pasting the text on the original figures. To improve these figures, we now pasted the text with a consistent font in the title.

All the best,
Uli Foelsche

---

## Author Response (AR4)

Response to the copy Editor:

We corrected the label error for the 2$^{nd}$ Fig.A1, which should be Fig.C1. In addition, we also renamed Table A1 to Table C1 for consistency.